# Batch Normalization Provably Avoids Rank Collapse for Randomly Initialised Deep Networks

**Hadi Daneshmand**[*]
INRIA Paris, ETH Zurich
seyed.daneshmand@inria.fr

**Jonas Kohler**[*]
Department of Computer Science, ETH Zurich
jonas.kohler@inf.ethz.ch

**Francis Bach**
INRIA-ENS-PSL, Paris
francis.bach@inria.fr

**Thomas Hofmann**
Department of Computer Science, ETH Zurich
thomas.hofmann@inf.ethz.ch

**Aurelien Lucchi**
Department of Computer Science, ETH Zurich
aurelien.lucchi@inf.ethz.ch

## Abstract

Randomly initialized neural networks are known to become harder to train with increasing depth, unless architectural enhancements like residual connections and batch normalization are used. We here investigate this phenomenon by revisiting the connection between random initialization in deep networks and spectral instabilities in products of random matrices. Given the rich literature on random matrices, it is not surprising to find that the rank of the intermediate representations in unnormalized networks collapses quickly with depth. In this work we highlight the fact that batch normalization is an effective strategy to avoid rank collapse for both linear and ReLU networks. Leveraging tools from Markov chain theory, we derive a meaningful lower rank bound in deep linear networks. Empirically, we also demonstrate that this rank robustness generalizes to ReLU nets. Finally, we conduct an extensive set of experiments on real-world data sets, which confirm that rank stability is indeed a crucial condition for training modern-day deep neural architectures.

## 1 Introduction and related work

Depth is known to play an important role in the expressive power of neural networks [28]. Yet, increased depth typically leads to a drastic slow down of learning with gradient-based methods, which is commonly attributed to unstable gradient norms in deep networks [15]. One key aspect of the training process concerns the way the layer weights are initialized. When training contemporary neural networks, both practitioners and theoreticians advocate the use of randomly initialized layer weights with i.i.d. entries from a zero mean (Gaussian or uniform) distribution. This initialization strategy is commonly scaled such that the variance of the layer activation stays constant across layers [13, 14]. However, this approach can not avoid spectral instabilities as the depth of the network increases. For example, [26] observes that for linear neural networks, such initialization lets all but one singular values of the last layers activation collapse towards zero as the depth increases.

Nevertheless, recent advances in neural architectures have allowed the training of very deep neural networks with standard i.i.d. initialization schemes *despite* the above mentioned shortcomings.

---

[*]Shared first authorship

Among these, both residual connections and normalization layers have proven particularly effective and are thus in widespread use (see [17, 24, 14] to name just a few). Our goal here is to bridge the explanatory gap between these two observations by studying the effect of architectural enhancements on the spectral properties of randomly initialized neural networks. We also provide evidence for a strong link of the latter with the performance of gradient-based optimization algorithms.

One particularly interesting architectural component of modern day neural networks is Batch Normalization (BN) [17]. This simple heuristics that normalizes the pre-activation of hidden units across a mini-batch, has proven tremendously effective when training deep neural networks with gradient-based methods. Yet, despite of its ubiquitous use and strong empirical benefits, the research community has not yet reached a broad consensus, when it comes to a theoretical explanation for its practical success. Recently, several alternatives to the original "internal covariate shift" hypothesis [17] have appeared in the literature: decoupling optimization of direction and length of the parameters [20], auto-tuning of the learning rate for stochastic gradient descent [3], widening the learning rate range [7], alleviating sharpness of the Fisher information matrix [18], and smoothing the optimization landscape [25]. Yet, most of these candidate justifications are still actively debated within the community. For example, [25] first made a strong empirical case against the original internal covariate shift hypothesis. Secondly, they argued that batch normalization simplifies optimization by smoothing the loss landscape. However, their analysis is on a per-layer basis and treats only the largest eigenvalue. Furthermore, even more recent empirical studies again dispute these findings, by observing the exact opposite behaviour of BN on a ResNet20 network [34].

## 1.1 On random initialization and gradient based training

In light of the above discussion, we take a step back – namely to the beginning of training – to find an interesting property that is provably present in batch normalized networks and can serve as a solid basis for a more complete theoretical understanding.

The difficulty of training randomly initialized, un-normalized deep networks with gradient methods is a long-known fact, that is commonly attributed to the so-called vanishing gradient effect, i.e., a decreasing gradient norm as the networks grow in depth (see, e.g., [27]). A more recent line of research tries to explain this effect by the condition number of the input-output Jacobian (see, e.g., [32, 33, 23, 7]). Here, we study the spectral properties of the above introduced initialization with a particular focus on the rank of the hidden layer activations over a batch of samples. The question at hand is whether or not the network preserves a diverse data representation which is necessary to disentangle the input in the final classification layer.

As a motivation, consider the results of Fig. 1, which plots accuracy and output rank when training batch-normalized and un-normalized neural networks of growing depth on the Fashion-MNIST dataset [31]. As can be seen, the rank in the last hidden layer of the vanilla networks collapses with depth and they are essentially unable to learn (in a limited number of epochs) as soon as the number of layers is above 10. The rank collapse indicates that the direction of the output vector has become independent of the actual input. In other words, the randomly initialized network no longer preserves information about the input. Batch-normalized networks, however, preserve a high rank across all network sizes and their training accuracy drops only very mildly as the networks reach depth 32.

The above example shows that both rank and optimization of even moderately-sized, unnormalized networks scale poorly with depth. Batch-normalization, however, stabilizes the rank in this setting and the obvious question is whether this effect is just a slow-down or even simply a numerical phenomenon, or whether it actually generalizes to networks of infinite depth.

In this work we make a strong case for the latter option by showing a remarkable stationarity aspect of BN. Consider for example the case of passing $N$ samples $x_i \in \mathbb{R}^d$ arranged column-wise in an input matrix $X \in \mathbb{R}^{d \times N}$ through a very deep network with fully-connected layers. Ideally, from an information propagation perspective, the network should be able to differentiate between individual samples, regardless of its depth [27]. However, as can be seen in Fig. 2, the hidden representation of $X$ collapses to a rank one matrix in vanilla networks, thus mapping all $x_i$ to the same line in $\mathbb{R}^d$. Hence, the hidden layer activations and along with it the individual gradient directions become

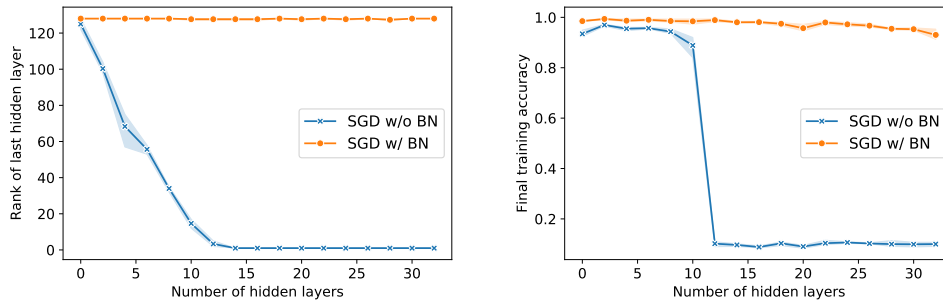

Figure 1: **Effect of depth on rank and learning**, on the Fashion-MNIST dataset with ReLU multilayer perceptrons (MLPs) of depth 1-32 and width 128 hidden units. Left: Rank[3] after random initialization as in PyTorch [22]. Right: Training accuracy after training 75 epochs with SGD, batch size 128 and grid-searched learning rate. Mean and 95% confidence interval of 5 independent runs.

independent from the input $x_i$ as depth goes to infinity. We call this effect "directional" gradient vanishing (see Section 3 for a more thorough explanation).

Interestingly this effect does not happen in batch-normalized networks, which yield – as we shall prove in Theorem 2 – a stable rank for *any* depth, thereby preserving a disentangled representation of the input and hence allowing the training of very deep networks. These results substantiate earlier empirical observations made by [7] for random BN-nets, and also validates the claim that BN helps with *deep information propagation* [27].

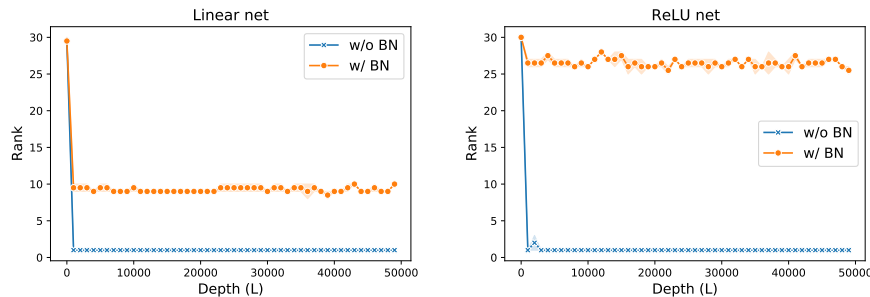

Figure 2: **Rank comparison of last hidden activation**: Log(rank) of the last hidden layer's activation over total number of layers (blue for BN- and orange for vanilla-networks) for Gaussian inputs. Networks are MLPs of width $d = 32$. (Left) Linear activations, (Right) ReLU activations. Mean and 95% confidence interval of 10 independent runs. While the rank quickly drops in depth for both networks, BN stabilizes the rank above $\sqrt{d}$.

## 1.2 Contributions

In summary, the work at hand makes the following two key contributions:

**(i)** We theoretically prove that BN indeed avoids rank collapse for deep linear neural nets under standard initialization and for any depth. In particular, we show that BN can be seen as a computationally cheap rank preservation operator, which may not yield hidden matrices with full rank but still preserves sufficient modes of variation in the data to achieve a scaling of the rank with $\Omega(\sqrt{d})$, where $d$ is the width of the network. Subsequently, we leverage existing results from random matrix theory [9] to complete the picture with a simple proof of the above observed rank collapse for linear vanilla networks, which interestingly holds regardless of the presence of residual connections (Lemma 3). Finally, we connect the rank to difficulties in gradient based training of deep nets by showing that rank collapse makes the directional component of the gradients independent of the input.

**(ii)** We empirically show that the rank is indeed a crucial quantity for gradient-based learning. In particular, we show that both the rank and the final training accuracy quickly diminish in depth unless

BN layers are incorporated in both simple feed-forward and convolutional neural nets. To take this reasoning beyond mere correlations, we actively intervene with the rank of networks before training and show that (a) one can break the training stability of BN by initializing in a way that reduces its rank-preserving properties, and (b) a rank-increasing pre-training procedure for vanilla networks can recover their training ability even for large depth. Interestingly, our pre-training method allows vanilla SGD to outperform BN on very deep MLPs. In all of our experiments, we find that SGD updates preserve the order of the initial rank throughout optimization, which underscores the importance of the rank at initialization for the entire convergence behavior.

## 2 Background and Preliminaries

**Network description.** We consider a given input $X \in \mathbb{R}^{d \times N}$ containing $N$ samples in $\mathbb{R}^d$. Let $\mathbf{1}_k \in \mathbb{R}^k$ denote the k-dimensional all one vector and $H_\ell^{(\gamma)}$ denote the hidden representation of $X$ in layer $\ell$ of a BN-network with residual connections. The following recurrence summarizes the network mapping

$$H_{\ell+1}^{(\gamma)} = \text{BN}_{0,\mathbf{1}_d}(H_\ell^{(\gamma)} + \gamma W_\ell H_\ell^{(\gamma)}), \quad H_0^{(\gamma)} = X, \tag{1}$$

where $W_\ell \in \mathbb{R}^{d \times d}$ and $\gamma$ regulates the skip connection strength (in the limit, $\gamma = \infty$ recovers a network without skip connection)[4]. Throughout this work, we consider the network weights $W_\ell$ to be initialized as follows.

**Definition 1** (Standard weight initialization). *The elements of weight matrices $W_\ell$ are i.i.d. samples from a distribution $\mathcal{P}$ that has zero-mean, unit-variance, and its density is symmetric around zero[5]. We use the notation $\mu$ for the probability distribution of the weight matrices.*

We define the BN operator $\text{BN}_{\alpha,\beta}$ as in the original paper [17], namely

$$\text{BN}_{\alpha,\beta}(H) = \beta \circ (\text{diag}\,(M(H)))^{-1/2} H + \alpha \mathbf{1}_N^\top, M(H) := \frac{1}{N} H H^\top, \tag{2}$$

where $\circ$ is a row-wise product. Both $\alpha \in \mathbb{R}^d$ and $\beta \in \mathbb{R}^d$ are trainable parameters. Throughout this work we assume the initialization $\alpha = 0$ and $\beta = \mathbf{1}_d$, and also omit corrections of the mean activity. As demonstrated empirically in Fig. 5, and theoretically in App. C this simplification does not change the performance of BN in our settings.

**Rank notions.** To circumvent numerical issues involved in rank computations we introduce a soft notion of the rank denoted by $\text{rank}_\tau(H)$ (soft rank). Specifically, let $\sigma_1, \ldots, \sigma_d$ be the singular values of $H$. Then, given a $\tau > 0$, we define $\text{rank}_\tau(H)$ as

$$\text{rank}_\tau(H) = \sum_{i=1}^d \mathbf{1}(\sigma_i^2/N \geq \tau). \tag{3}$$

Intuitively, $\text{rank}_\tau(H)$ indicates the number of singular values whose absolute values are greater than $\sqrt{N\tau}$. It is clear that $\text{rank}_\tau(H)$ is less or equal to $\text{rank}(H)$ for all matrices $H$. For analysis purposes, we need an analytic measure of the collinearity of the columns and rows of $H$. Inspired by the so-called stable rank (see, e.g., [29]), we thus introduce the following quantity

$$r(H) = \text{Tr}(M(H))^2/\|M(H)\|_F^2, \quad M(H) = HH^\top/N. \tag{4}$$

In contrast to the algebraic rank, $r(H)$ is differentiable with respect to $H$. Furthermore, the next lemma proves that the above quantity lower-bounds the soft-rank for the hidden representations.

**Lemma 1.** *For an arbitrary matrix $H \in \mathbb{R}^{d \times d}$, $\text{rank}(H) \geq r(H)$. For the sequence $\{H_\ell^{(\gamma)}\}_{\ell=1}^\infty$ defined in Eq. (2), $\text{rank}_\tau(H_\ell^{(\gamma)}) \geq (1-\tau)^2 r(H_\ell^{(\gamma)})$ holds for $\tau \in [0,1]$.*

# 3 Batch normalization provably prevents rank collapse

Since our empirical observations hold equally for both non-linear and linear networks, we here focus on improving the theoretical understanding in the linear case, which constitutes a growing area of research [26, 19, 6, 2]. First, inspired by [10] and leveraging tools from Markov Chain theory, our main result proves that the rank of linear batch-normalized networks scales with their width as $\Omega(\sqrt{\text{width}})$. Secondly, we leverage results from random matrix theory [8] to contrast our main result to unnormalized linear networks which we show to provably collapse to rank one, even in the presence of residual connections.

## 3.1 Main result

In the following we state our main result which proves that batch normalization indeed prevents the rank of all hidden layer activations from collapsing to one. Please see Appendix E for the more formal version of this theorem statement.

**Theorem 2.** *[Informal] Suppose that the rank$(X) = d$ and that the weights $W_\ell$ are initialized in a standard i.i.d. zero-mean fashion (see Def. 1). Then, the following limits exist such that*

$$\lim_{L \to \infty} \frac{1}{L} \sum_{\ell=1}^{L} rank_\tau(H_\ell^{(\gamma)}) \geq \lim_{L \to \infty} \frac{(1-\tau)^2}{L} \sum_{\ell=1}^{L} r(H_\ell^{(\gamma)}) = \Omega((1-\tau)^2 \sqrt{d}) \tag{5}$$

*holds almost surely for a sufficiently small $\gamma$ (independent of $\ell$) and any $\tau \in [0, 1)$, under some additional technical assumptions. Please see Theorem 14 in the Appendix for the formal statement.*

Theorem 2 yields a non trivial width-dependency. Namely, by setting for example $\tau := 1/2$, the result states that the average number of singular values with absolute value greater than $\sqrt{N/2}$ is at least $\Omega(\sqrt{d})$ on average. To put this into context: If one were to replace $\text{diag}(M)^{-1/2}$ by the *full* inverse $(M)^{-1/2}$ in Eq. (2), then BN would effectively constitute a classical whitening operation such that all $\{H_\ell^{(\gamma)}\}_{\ell=1}^{L}$ would be full rank (equal to $d$). However, as noted in the original BN paper [17], whitening is obviously expensive to compute and furthermore prohibitively costly to incorporate in back-propagation. As such, BN can be seen as a computationally inexpensive approximation of whitening, which does not yield full rank hidden matrices but still preserves sufficient variation in the data to provide a rank scaling as $\Omega(\sqrt{d})$. Although the lower-bound in Thm. 2 is established on the average over infinite depth (i.e., $L \to \infty$), Corollary 15 (in App. E) proves that the same bound holds for all rank$(H_\ell)$ and rank$_\tau(H_\ell)$.

**Necessary assumptions.** The above result relies on two key assumptions: (i) First, the input $X$ needs to be full rank. (ii) Second, the weights have to be drawn according to the standard initialization scheme. We believe that both assumptions are indeed necessary for BN to yield a robust rank.

Regarding (i), we consider a high input rank a natural condition since linear neural nets cannot possibly *increase* the rank when propagating information through their layers. Of course, full rank is easily achieved by an appropriate data pre-processing. Yet, even when the matrix is close to low rank we find that BN is actually able to amplify small variations in the data (see Fig. 3.b).[6] Notably, we observed that hidden representations remain full rank if $H_0^{(\gamma)}$ is full-rank and $N = O(\sqrt{d})$. Regarding (ii), we derive – based on our theoretical insights – an adversarial initialization strategy that corrupts both the rank robustness and optimization performance of batch-normalized networks, thus suggesting that the success of BN indeed relies heavily on the standard i.i.d. zero-mean initialization.

**Experimental validation.** In order to underline the validity of Theorem 2 we run multiple simulations by feeding Gaussian data of dimensionality $d = N$ into networks of growing size and with different residual strengths. For each network, we compute the mean and standard deviation of the soft rank rank$_\tau$ with $\tau = 0.5$. As depicted in Fig. 3, the curves clearly indicate a $\Omega(\sqrt{d})$ dependency for $\lim_{L \to \infty} \sum_{\ell=1}^{L} \text{rank}_\tau(H_\ell)/L$, just as predicted in the Theorem. Although the established guarantee requires the weight on the parametric branch (i.e., $\gamma$) to be small, the results of Fig. 3 indicate that

the established lower bound holds for a much wider range including the case where no residual connections are used at all ($\gamma = \infty$).

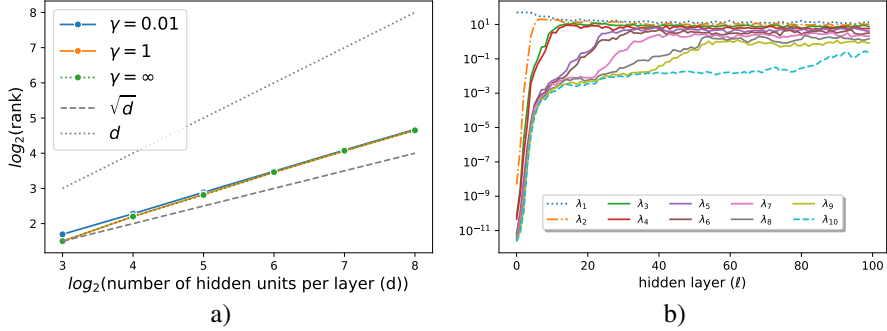

a)                                                                                              b)

Figure 3: a) Result of Theorem 2 for different values of $\gamma$, where $\gamma = \infty$ depicts networks *without* skip connections. Each point is the average rank$_{1/2}$ over depth ($L = 10^6$) of nets of width $d \in \{8, 16, .., 256\}$ an on x-axis. b) Top 10 singular values of $H_\ell^{(\gamma)}$ for increasing values of $\ell$ given nearly collinear inputs. As can be seen, BN quickly amplifies smaller variations in the data while reducing the largest one.

## 3.2 Comparison with unnormalized networks

In order to stress the importance of the above result, we now compare the predicted rank of $H_\ell$ with the rank of unnormalized linear networks, which essentially constitute a linear mapping in the form of a product of random matrices. The spectral distribution of products of random matrices with i.i.d. standard Gaussian elements has been studied extensively [7, 12, 21]. Interestingly, one can show that the gap between the top and the second largest singular value increases with the number of products (i.e., $\ell$) at an exponential rate[7] [12, 21]. Hence, the matrix converges to a rank one matrix after normalizing by the norm. In the following, we extend this result to products of random matrices with a residual branch that is obtained by adding the identity matrices. Particularly, we consider the hidden states $\widehat{H}_\ell$ of the following linear residual network:

$$\widehat{H}_\ell = \mathbf{B}_\ell X, \quad \mathbf{B}_\ell := \prod_{k=1}^{\ell}(I + \gamma W_k). \tag{6}$$

Since the norm of $\widehat{H}_\ell$ is not necessarily bounded, we normalize as $\widetilde{H}_\ell = B_\ell X / \|B_\ell\|$. The next lemma characterizes the limit behaviour of $\{\widetilde{H}_\ell\}$.

**Lemma 3.** *Suppose that $\gamma \in (0, 1)$ and assume the weights $W_\ell$ to be initialized as in Def. 1 with element-wise distribution $\mathcal{P}$. Then we have for linear networks, which follow recursion (6), that:*

a. *If $\mathcal{P}$ is standard Gaussian, then the sequence $\{\widetilde{H}_\ell\}$ converges to a rank one matrix.*

b. *If $\mathcal{P}$ is uniform$[-\sqrt{3}, \sqrt{3}]$, then there exists a monotonically increasing sequence of integers $\ell_1 < \ell_2, \ldots$ such that the sequence $\{\widetilde{H}_{\ell_k}\}$ converges to a rank one matrix.*

This results stands in striking contrast to the result of Theorem 2 established for batch-normalized networks.[8] Interestingly, even residual skip connections cannot avoid rank collapse for very deep neural networks, unless one is willing to incorporate a depth dependent down-scaling of the parametric branch as for example done in [1], who set $\gamma = O(\frac{1}{L})$ . Remarkably, Theorem 2 shows that BN layers provably avoid rank collapse *without* requiring the networks to become closer and closer to identity. Remarkably, the remaining direction after rank collapse depends exclusively on the random weights and it is independent of the input.

**Implications of rank collapse on gradient based learning.** In order to explain the severe conse-quence of rank collapse on optimization performance reported in Fig. 1, we study the effect of rank one hidden-layer representations on the gradient of the training loss for distinct input samples. Let $\mathcal{L}_i$ denote the training loss for datapoint $i$ on a vanilla network as in Eq. (6). Furthermore, let the final classification layer be parametrized by $W_{L+1} \in \mathbb{R}^{d_{out} \times d}$. Then, given that the hidden presentation at the last hidden layer $L$ is rank one, the normalized gradients of the loss with respect to weights of individual neurons $k \in 1, ..., d_{out}$ in the classification layer (denoted by $\nabla_{W_{L+1,k}} \mathcal{L}_i$, where $\|\nabla_{W_{L+1,k}} \mathcal{L}_i\| = 1$) are collinear for any two datapoints $i$ and $j$, i.e. $\nabla_{W_{L+1,k}} \mathcal{L}_i = \mp \nabla_{W_{L+1,k}} \mathcal{L}_j$. A formal statement is presented in Prop. 19 in the Appendix alongside empirical validations on a VGG19 network (Fig. 10). This result implies that the commonly accepted vanishing gradient *norm* hypothesis is not descriptive enough since SGD does not take small steps into the *right* direction, but into a random direction that is *independent* from the input. In other words, deep neural networks are prone to *directional gradient vanishing* after initialization, which is caused by the collapse of the last hidden layer activations to a very small subspace (one line in $\mathbb{R}^d$ in the extreme case of rank one activations).

# 4   The important role of the rank

The preceding sections highlight that the rank of the hidden representations is a key difference between random vanilla and BN networks. We now provide three experimental findings that substantiate the particular importance of the rank at the beginning of training: First, we find that an unsupervised, rank-increasing pre-training allows SGD on vanilla networks to outperform BN networks. Second, we show that the performance of BN-networks is closely tied to a high rank at initialization. Third, we report that SGD updates preserve the initial rank magnitude throughout the optimization process.

**Outperforming BN using a pre-training step.** As discussed above, batch normalization layers are very effective at avoiding rank collapse. Yet, this is of course not the only way to preserve rank. Based upon our theoretical insights, we leverage the lower bound established in Eq. (4) to design a pre-training step that not only avoids rank collapse but also accelerates the convergence of SGD. Our proposed procedure is both simple and computationally cheap. Specifically, we *maximize* the lower-bound $r(H_\ell)$ (in Eq. (4)) on the rank of the hidden presentation $H_\ell$ in each layer $\ell$. Since this function is differentiable with respect to its input, it can be optimized sufficiently by just a few steps of (stochastic) gradient ascent (see Section G in the Appendix for more details).

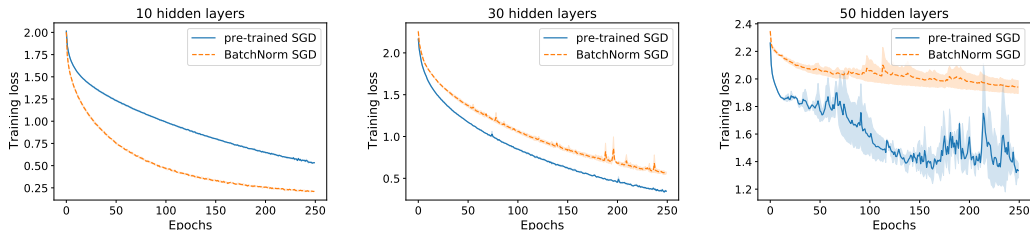

Figure 4: **Pre-training versus** BN**:** Loss over epochs on CIFAR-10 for MLPs of increasing depth with 128 hidden units and ReLU activation. Trained with SGD (batchsize 64) and grid-searched stepsize. See Fig. 11 for the corresponding test loss and accuracy as well as Fig. 12 for FashionMNIST results.

Fig. 4 compares the convergence rate of SGD on pre-trained vanilla networks and BN-networks. As can be seen, the slow down in depth is much less severe for the pre-trained networks. This improvement is, also, reflected both in terms of training accuracy and test loss (see Fig. 11 in Appendix). Interestingly, the pre-training is not only faster than BN on deep networks, but it is also straight-forward to use in settings where the application of BN is rather cumbersome such as for very small batch sizes or on unseen data [16, 30].

**Breaking batch normalization.** Some scholars hypothesize that the effectiveness of BN stems from a global landscape smoothing [25] or a certain learning rate tuning [3], that are thought to be induced by the normalization. Under these hypotheses, one would expect that SGD converges fast on BN-nets *regardless* of the initialization. Yet, we here show that the way that networks are initialized does play a crucial role for the subsequent optimization performance of BN-nets.

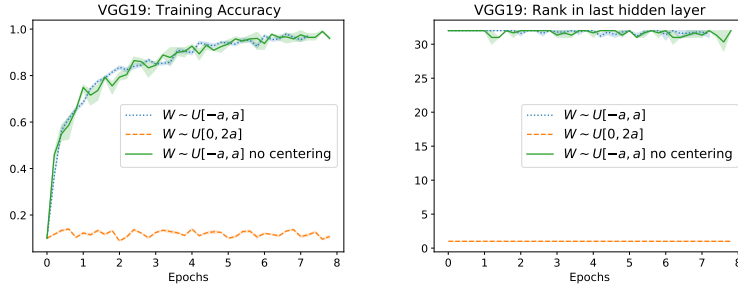

Figure 5: **Breaking Batchnorm:** CIFAR-10 on VGG19 with standard PyTorch initialization as well as a uniform initialization of same variance. (Left) training accuracy, (Right) Rank of last hidden layer computed using $torch.matrix\_rank()$. Plot also shows results for standard initialization and BN *without* mean deduction. Avg. and 95% CI of 5 independent runs. (See Fig. 13 in Appendix for similar results on ResNet-50).

Particularly, we train two MLPs with batchnorm, but change the initialization for the second net from the standard PyTorch way $W_{l,i,j} \sim$ uniform $\left[-1/\sqrt{d_l}, 1/\sqrt{d_l}\right]$ [22, 13] to $W_{l,i,j} \sim$ uniform $\left[0, +2/\sqrt{d_l}\right]$, where $d_l$ is the layer size. As can be seen to the right, this small change reduces the rank preserving quality of BN significantly, which is reflected in much slower learning behaviour. Even sophisticated modern day architectures such as VGG and ResNet networks are unable to fit the CIFAR-10 dataset after changing the initialization in this way (see Fig. 5).

**Rank through the optimization process.** The theoretical result of Theorem 2 considers the rank at random initialization. To conclude, we perform two further experiments which confirm that the initial rank strongly influences the speed of SGD throughout the entire optimization process. In this regard, Fig. 6 reports that SGD preserves the initial magnitude of the rank to a large extent, regardless of the specific network type. This is particularly obvious when comparing the two BN initializations. A further noteworthy aspect is the clear correlation between the level of pre-training and optimization performance on vanilla nets. Interestingly, this result does again not only hold on simple MLPs but also generalizes to modern day networks such as the VGG-19 (see Fig. 5) and ResNet50 architecture (see Appendix I).

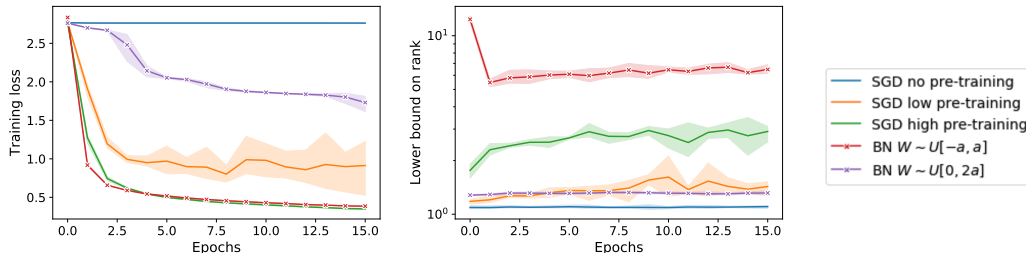

Figure 6: **Pretraining:** Fashion-MNIST on MLPs of depth 32 and width 128. (Left) Training accuracy, (Right) Lower bound on rank. Blue line is a ReLU network with standard initialization. Other solid lines are pre-trained layer-wise with 25 (orange) and 75 (green) iterations to increase the rank. Dashed lines are batchnorm networks with standard and asymmetric initialization. Average and 95% confidence interval of 5 independent runs.

## 5   Discussions

In summary, our work highlights a key difference between random vanilla- and BN networks. While the rank of the hidden representations quickly collapses to one as the depth of vanilla networks increases, BN is robust against such rank collapse. This intriguing property arises due to the standard initialization of weights and also it is preserved through the optimization process. Notably, our theoretical analysis proves this striking difference for linear MLPs and holds empirically across a wide range of data sets and network architectures. Our experiments further highlight the determining

role of the rank quantity in the training speed. Inspired by these observations, we develop a novel pre-training method that allows previously un-trainable very deep vanilla networks to learn, sometimes even faster than BN-MLPs of the same size. Thereby our study reveals a key requirement for a proper initialization of deep neural networks, opening doors to the development of effective initialization schemes for modern-day architectures.

We thus consider our work a relevant step towards a better understanding of optimization for deep neural networks. Furthermore, our findings give rise to several interesting follow-up questions: (i) Can one generalize the analysis of Theorem 2 to ReLU and other non-linear nets to prove the observed rank robustness (e.g. Fig. 2)? (ii) is it possible to rigorously prove that SGD updates preserve the rank magnitude throughout optimization, as observed in Fig. 6)? (iii) Is it possible to use the develop a similarly effective pre-training for convolution and recurrent networks? (iv) How can one theoretically characterize the connection between the convergence of SGD and the rank quantity (a follow-up on directional gradient vanishing)? (v) Does rank robustness explain the success of related architectures such as layer normalization [4], weight normalization [24]) and modern initialization techniques such as fix up initialization [35]? We believe that these questions will spark an interesting line of future research towards the goal of fully understanding optimization in deep neural networks.

## Broader impact

As we only contribute to a better understanding of neural network training in general, we consider our work fundamental research without any specific application. Hence a broader impact discussion is not applicable.

## Funding

This project is fully supported by ETH-Zurich fellowships.

## Acknowledgement

We thank Dr. Aran Raoufi, Dr. Olivier Ledoit, Gary Becigneul, and Olivier Ledoit for their helpful discussions.

## Footnotes

[3]Computed using $torch.matrix\_rank()$, which regards singular values below $\sigma_{\max} \times d \times 10^{-7}$ as zero. This is consistent with both Matlab and Numpy.

[4]For the sake of simplicity, we here assume that the numbers of hidden units is equal across layers. In App. E we show how our results extend to nets with varying numbers of hidden units.

[5]Two popular choices for $\mathcal{P}$ are the Gaussian distribution $\mathcal{N}(0,1)$ and the uniform distribution $\mathcal{U}([-1,1])$. The variance can be scaled with the choice of $\gamma$ to match the prominent initializations from [14] and [13]. Note that the symmetry implies that the law of each element $[W_\ell]_{ij}$ equates the law of $-[W_\ell]_{ij}$.

[6]Intuitively this means that even if two data points are very close to each other in the input space, their hidden presentation can still be disentangled in batch-normalized networks (see Appendix E for more details)

[7]The growth-rate of the $i$-th singular value is determined by the $i$-th Lyapunov exponent of the product of random matrices. We refer the reader to [12] for more details on Lyapunov exponents.

[8]According to the observations in Fig. 2, the result of part b holds for the usual sequence of indices $\{\ell_k = k\}$, which indicates that $\{\widetilde{H}_k\}$ converges to a rank one matrix even in the case of uniform initialization.

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
