[Supplementary Material]

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

[9]When $d$ is sufficiently large and assuming that coordinates in one row are weakly dependent, the central limit theorem implies that the empirical average of the rows converges to zero.

[10] Recall that the updates in Eq. (1) is obtained by matrix multiplications, hence it does not increase the rank.

[11]Recall $\text{Tr}(M_\ell) = d$ holds due to property (p.2) in Eq. 9

[12]The uniqueness of the invariant distribution implies Ergodicity (see Theorem 5.2.1 and 5.2.6 [11]).

[13]According to definition, $\lim_{d\to\infty} o(d^\beta)/\Theta(d^\beta) = 0$

[14]This result is obtained by setting $g(H) = \|M(H)\|_F^2$ in Def. 2.

[15]Note that $H \in \mathcal{H}$ is a random matrix whose law is $\nu_\gamma$, hence $\lambda \in \mathbb{R}^d$ is also a random vector.

[16]Recall that the rank does not increases in updates of Eq. (7)

[17]Notably, the absolute value of each element of $\frac{1}{d} \sum_{i=1}^{d} \alpha_i e_i v^\top$ is less than 1, hence this matrix belongs to the support of $\mu$.

[18]A single line in $\mathbb{R}^d$ in the extreme case of rank one mappings

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

# Appendix

## A   Preliminaries

Recall that $H_\ell^{(\gamma)}$ denotes the hidden representations in layer $\ell$. These matrices make a Markov chain that obeys the recurrence of Eq. (1), which we restate here

$$H_{\ell+1}^{(\gamma)} = \mathrm{BN}(H_\ell^{(\gamma)} + \gamma W_\ell H_\ell^{(\gamma)}), \quad H_0^\gamma = X, \tag{7}$$

where we use the compact notation BN for $\mathrm{BN}_{0,\mathbf{1}_d}$. Let $M_\ell^{(\gamma)}$ be second moment matrix of the hidden representations $H_\ell^{(\gamma)}$, i.e. $M_\ell^{(\gamma)} := H_\ell^{(\gamma)} \left( H_\ell^{(\gamma)} \right)^\top /N$. Batch normalization ensures that the rows of $H_\ell$ have the same norm $\sqrt{N}$ for $\ell > 0$ –where $N$ is the size of mini-batch. Let $\mathcal{H}$ be space of $d \times d$-matrices that obey this propery. This property enforces two key characteristics on $M_\ell^{(\gamma)}$:

(p.1)     its diagonal elements are one                                             (8)

(p.2)     the absolute value of its off-diagonal elements is less than one          (9)

Property (p.1) directly yields that the trace of $M_\ell^{(\gamma)}$ (and hence the sum of its eigenvalues) is equal to $d$. We will repeatedly use these properties in our analysis.

Furthermore, the sequence $\{H_\ell^{(\gamma)}\}_{\ell=1}^\infty$ constitute a Markov chain. Under mild assumptions, this chain admits an invariant distribution that is defined bellow[11].

**Definition 2.** *Distribution $\nu$ is an invariant distribution of the hidden representations $\{H_\ell^{(\gamma)}\}_{\ell=1}^\infty$ if it obeys*

$$\int \mathrm{BN}(H + \gamma W H)\mu(dW)\nu(dH) = \int \mathrm{BN}(H)\nu(dH) \tag{10}$$

*where $\mu$ denotes the probability measure of random weights.*

Later, we will see that the above invariance property allows us to determine the commutative behaviour of the sequence of hidden presentations.

## B   Lower bounds on the (soft) rank

Recall that we introduced the ratio $r(H) = \mathrm{Tr}(M(H))^2 / \|M(H)\|_F^2$ in Eq. (4) as a lower bound on both the $\mathrm{rank}(H)$ as well as the soft rank $\mathrm{rank}_\tau(H)$ (stated in Lemma 1). This section establishes these lower bounds.

*Proof of Lemma 1 (part 1).* We first prove that $\mathrm{rank}(H) \geq r(H)$. Let $M := M(H) = HH^\top/N$. Since the eigenvalues of $H$ are obtained by a constant scaling factor of the squared singular values of $H$, these two matrices have the same rank. We now establish a lower bound on $\mathrm{rank}(M)$. Let $\lambda \in \mathbb{R}^d$ contain the eigenvalues of matrix $M$, hence $\|\lambda\|_1 = \mathrm{Tr}(M)$ and $\|\lambda\|_2^2 = \|M\|_F^2$. Given $\lambda$, we define the vector $w \in \mathbb{R}^d$ as

$$w_i = \begin{cases} 1/\|\lambda\|_0 & : \lambda_i \neq 0 \\ 0 & : \lambda_i = 0. \end{cases} \tag{11}$$

To proof the assertion, we make use if a straightforward application of Cauchy-Schwartz

$$|\langle \lambda, w \rangle| \leq \|\lambda\|_2 \|w\|_2 \tag{12}$$

$$\implies \|\lambda\|_1/\|\lambda\|_0 \leq \|\lambda\|_2/\|\lambda\|_0^{1/2} \tag{13}$$

$$\implies \frac{\|\lambda\|_1}{\|\lambda\|_2} \leq \|\lambda\|_0^{1/2}. \tag{14}$$

Replacing $\|\lambda\|_2 = \|M\|_F$ and $\|\lambda\|_1 = \mathrm{Tr}(M)$ into the above equation concludes the result. Note that the above proof technique has been used in the planted sparse vector problem [5]. $\square$

*Proof of Lemma 1 (part 2).* Now, we prove that $\operatorname{rank}_\tau(H_\ell^{(\gamma)}) \geq (1-\tau)^2 r(H_\ell^{(\gamma)})$. Let $\lambda \in \mathbb{R}_+^d$ be a vector containing the eigenvalues of the matrix $M_\ell^{(\gamma)} = M(H_\ell^{(\gamma)})$. Let $\sigma \in \mathbb{R}_+^d$ contain the singular values of $H$. Then, one can readily check that $\sigma_i^2/N = \lambda_i$. Furthermore, $\|\lambda\|_1 = d$ due to (p.1) in Eq. (8). Furthermore, we have by definition that

$$\operatorname{rank}_\tau(H_\ell^{(\gamma)}) = h_\tau(\lambda) := \sum_{i=1}^d \mathbf{1}(\sigma_i^2/N \geq \tau) = \sum_{i=1}^d \mathbf{1}(\lambda_i \geq \tau). \tag{15}$$

Let us now define a vector $w \in \mathbb{R}^d$ with entries

$$w_i = \begin{cases} 1/h_\tau(\lambda) & : \lambda_i \geq \tau \\ 0 & : \text{otherwise.} \end{cases} \tag{16}$$

Then, we use Cauchy-Schwartz to get

$$|\langle \lambda, w \rangle| \leq \|\lambda\|_2 \|w\|_2. \tag{17}$$

It is easy to check that $\|w\|_2 = h_\tau(\lambda)^{-1/2}$ holds. Furthermore,

$$h_\tau(\lambda)|\langle w, \lambda \rangle| = \sum_{|\lambda_i| \geq \tau}^d |\lambda_i| \tag{18}$$

$$\geq \|\lambda\|_1 - d\tau \tag{19}$$

$$\geq (1-\tau)\|\lambda\|_1, \tag{20}$$

where we used the fact that $\|\lambda\|_1 = d$ in the last inequality. Replacing this result into the bound of Eq. (17) yields

$$\operatorname{rank}_\tau(H_\ell^{(\gamma)}) = h_\tau(\lambda) \geq (1-\tau)^2 \|\lambda\|_1^2/\|\lambda\|_2^2 = (1-\tau)^2 r(H_\ell^{(\gamma)}), \tag{21}$$

which conludes the proof. $\qquad\square$

## C  Initialization consequences

The particular weight initialization scheme consider through out this work (recall Def. 1), imposes an interesting structure in the invariant distribution of the sequence of hidden presentations (defined in Def. 2).

**Lemma 4.** *Suppose that the chain $\{H_\ell^{(\gamma)}\}_{\ell=1}^\infty$ (defined in Eq. 7) admits a unique invariant distribution $\nu_\gamma$ and $H$ is drawn from $\nu_\gamma$; then, the law of $H_{i:}$ equates the law of $-H_{i:}$ where $H_{i:}$ denotes the ith row of matrix $H$.*

*Proof.* Let $S$ be a sign filliping matrix: it is diagonal and its diagonal elements are in $\{+1, -1\}$. Then $SW \overset{d}{=} W$ holds for a random matrix $W$ whose elements are drawn i.i.d. from a symmetric distribution. Let $H$ be drawn from the invariant distribution of the chain denoted by $\nu_\gamma$; Leveraging the invariance property, we get

$$H \overset{d}{=} H_+ \overset{d}{=} \left(\operatorname{diag}(H_{1/2}H_{1/2}^\top/N)\right)^{-1/2} H_{1/2}, \quad H_{1/2} := H + \gamma S W S H \tag{22}$$

By multiplying both sides with $S$, we get

$$SH \overset{d}{=} SH_+ \overset{d}{=} \left(\operatorname{diag}\left(H_{1/2}H_{1/2}^\top/N\right)\right)^{-1/2} \widetilde{H}_{1/2}, \quad \widetilde{H}_{1/2} := SH + \gamma W S H \tag{23}$$

Note that we use the fact that diagonal matrices commute in the above derivation. According to the definition, $S^2 = I$ holds. Considering this fact, we get

$$\operatorname{diag}\left(H_{1/2}H_{1/2}^\top\right) = \operatorname{diag}\left((H + \gamma SWSH)(H + \gamma SWSH)^\top\right) \tag{24}$$

$$= \operatorname{diag}\left((SSH + \gamma SWSH)(SSH + \gamma SWSH)^\top\right) \tag{25}$$

$$= \operatorname{diag}\left(S(SH + \gamma WSH)(SH + \gamma WSH)^\top S\right) \tag{26}$$

$$= \operatorname{diag}\left((SH + \gamma WSH)(SH + \gamma WSH)^\top\right) \tag{27}$$

$$= \widetilde{H}_{1/2}\widetilde{H}_{1/2}^\top \tag{28}$$

Replacing the above result into Eq. (29) yields

$$SH \overset{d}{=} SH_+ \overset{d}{=} \text{diag}^{-1/2} \left( \widetilde{H}_{1/2} \widetilde{H}_{1/2}^\top / N \right) \widetilde{H}_{1/2}, \quad \widetilde{H}_{1/2} := SH + \gamma WSH. \quad (29)$$

Hence the law of $SH$ is invariant too. Since the invariant distribution is assumed to be unique, $SH \overset{d}{=} H$ holds and thus $H_{i:} \overset{d}{=} -H_{i:}$.

$\square$

**Comment on BN-centering**   Let $\nu_\gamma$ be the unique invariant distribution associated with Markov chain $\{H_\ell^{(\gamma)}\}$. A straightforward implication of last Lemma is $\mathbb{E}[H_i] = 0$ for $H \sim \nu_\gamma$, hence the rows of $H_\ell^{(\gamma)}$ are mean-zero, hence their average is close to zero [9] and the mean-zero operation in BN is redundant. Although this theoretical argument is established for linear networks, we empirically observed that BN without centering also works well on modern neural architectures. For example, Fig. 7 shows that the centering does not affect the performance of BN on a VGG net when training CIFAR-10.

Training Accuracy               $r(H_L)$

Figure 7: Centering for BN. The experiment is conducted on a VGG network. The blue line indicates the original BN network and the orange line is BN without mean adaption. The vertical axis in the left plot is training accuracy. In the right plot it is $r(H_L)$, where $H_L$ is the data representation in the last hidden layer $L$. The horizontal axis indicates the number of iterations.

# D   Main Theorem: warm-up analysis

As a warm-up analysis, the next lemma proves that $\text{rank}(H_\ell^{(\gamma)}) \geq 2$ holds. Later, we will prove a stronger result. Yet, this initial results provides valuable insights into our proof technique. Furthermore, we will use the following result in the next steps.

**Lemma 5.** *Suppose that each element of the weight matrices is independently drawn from distribution $\mathcal{P}$ that is zero-mean, unit-variance, and its support lies in interval $[-B, B]$. If the Markov chain $\{H_\ell\}_{\ell \geq 1}$ admits a unique invariant distribution, then*

$$\text{rank}(H_\ell^{(\gamma)}) \geq 2 \quad (30)$$

*holds almost surely for all integers $\ell$ and $\gamma \leq 1/(8d)$.*

*Proof.* Let the weights $\{W_\ell\}$ be drawn from the distribution $\mu$, defined in Def. 1. Such a distribution obeys an important property: element-wise symmetricity. That is, $[W_\ell]_{ij}$ is distributed as $-[W_\ell]_{ij}$. Such an initialization enforces an interesting structural property for the invariant distribution $\nu_\gamma$ that is stated in Lemma 4. It is easy to check that this implies

$$\mathbb{E}\left[ [M(H_\ell^{(\gamma)})]_{ij} \right] = -\mathbb{E}\left[ [M(H_\ell^{(\gamma)})]_{ij} \right] = 0, \quad (31)$$

for any $i \neq j$. Recall, $M(H) = HH^\top / N$. The above property enforces $[M(H)]_{ij}^2$ to be small and hence $\|M_\ell^{(\gamma)}\|_F^2$ is small as well. Now, as $\text{rank}(H_\ell^{(\gamma)})$ is proportional to $1/\|M_\ell^{(\gamma)}\|_F^2$ (compare

Eq. (4)), it must consequently stay large. The rest of the proof is based on this intuition. Given the uniqueness of the invariant distribution, we can invoke Birkhoff's Ergodic Theorem for Markov Chains (Theorem 5.2.1 and 5.2.6 [11]) which yields

$$\lim_{L \to \infty} \frac{1}{L} \sum_{\ell=1}^{L} [M_\ell^{(\gamma)}]_{ij} = \mathbb{E}_{H \sim \nu_\gamma} \left[ [M(H)]_{ij} \right]. \tag{32}$$

This allows us to conclude the proof by a simple contradiction. Assume that $\text{rank}(H_k^{(\gamma)})$ is indeed one. Then, as established in the following Lemma, in the limit all entries of $M(H_\ell^{(\gamma)})$ are constant and either $-1$ or $1$.

**Lemma 6.** *Suppose the assumptions of Lemma 5 hold. If $\text{rank}(H_k^{(\gamma)}) = 1$ for an integer $k$, then $M(H_\ell^{(\gamma)}) = M(H_k^{(\gamma)})$ holds for all $\ell > k$. Furthermore, all elements of all matrices $\{M(H_\ell^{(\gamma)})\}_{\ell \geq k}$ have absolute value one, hence*

$$\lim_{L \to \infty} \frac{1}{L} \sum_{\ell=1}^{L} [M(H_\ell^{(\gamma)})]_{ij} \in \{1, -1\} \tag{33}$$

*holds.*

As a result, leveraging the ergodicity established in (61), we get that then

$$\mathbb{E}_{H \sim \nu_\gamma} \left[ [M(H)]_{ij} \right] \in \{+1, -1\} \tag{34}$$

must also hold. However, this contradicts the consequence of the symmetricity (Eq. (31)) which states that for any $j \neq i$ we have $\mathbb{E}_{H \sim \nu_\gamma} \left[ [M(H)]_{ij} \right] = -\mathbb{E}_{H \sim \nu_\gamma} \left[ [M(H)]_{ij} \right] = 0$. Thus, the rank one assumption cannot hold, which proves the assertion. □

To complete the proof of the last theorem, we prove Lemma 6.

*Proof of Lemma 6.* For the sake of simplicity, we omit all superscripts $(\gamma)$ throughout the proof. Suppose that $\text{rank}(H_k) = 1$, then $\text{rank}(H_\ell) = 1$ for all $\ell \geq k$ as the sequence $\{\text{rank}(H_\ell)\}$ is non-increasing [10]. Invoking the established rank bound from Lemma 1, we get

$$r(H_\ell) = \frac{\text{Tr}(M_\ell)^2}{\|M_\ell\|_F^2} \leq \text{rank}(H_\ell) = 1. \tag{35}$$

The above inequality together with properties (p.1) and (p.2) (presented in Eqs 8 and 9) yield $\text{Tr}(M_\ell) = d$. Replacing this into the above equation gives that $\|M_\ell\|_F^2 \geq d^2$ must hold for the rank of $H_\ell$ to be one. Yet, recalling property (p.2), this can only be the case if $[M_\ell]_{ij} \in \{+1, -1\}$ for all $i, j$. Replacing the definition $M(H) = HH^\top/N$ into updates of hidden presentation in Eq. 1 obtains

$$M_{\ell+1} = \text{diag}\left(M_{\ell+\frac{1}{2}}\right)^{-1/2} \left(M_{\ell+\frac{1}{2}}\right) \text{diag}\left(M_{\ell+\frac{1}{2}}\right)^{-1/2} \tag{36}$$

where

$$M_{\ell+\frac{1}{2}} = M_\ell + \Delta M_\ell, \quad \Delta M_\ell := \gamma W_\ell M_\ell + \gamma M_\ell W_\ell^\top + \gamma^2 W_\ell M_\ell W_\ell^\top \tag{37}$$

We now prove that the sign of $[M_\ell]_{ij}$ and $[M_{\ell+1}]_{ij}$ are the same for $[M_\ell]_{ij} \in \{+1, -1\}$. The above update formula implies that the sign of $[M_{\ell+1}]_{ij}$ equates that of $[M_{\ell+1/2}]_{ij}$. Furthermore, it is easy to check that $|[\Delta M_\ell]_{ij}| \leq 4\gamma B$. For $\gamma \leq 1/(8Bd)$, this bound yields $|[\Delta M_\ell]_{ij}| \leq \frac{1}{2}$. Therefore, the sign of $[M_{\ell+1/2}]_{ij}$ is equal to the one of $[M_\ell]_{ij}$. Since furthermore $[M_{\ell+1}]_{ij} \in \{1, -1\}$ holds, we conclude that all elements of $M_\ell$ remain constant for all $\ell \geq k$, which yields the limit stated in Eq. 33 . □

# E  Main theorem: Proof

In this section, we prove that BN yields an $\Omega(\sqrt{d})$-rank for hidden representation.

*Proof sketch for Thm. 2.* The proof is based on an application of ergodic theory (as detailed for example in Section 5 of [11]). In fact, the chain of hidden representations, denoted by $H_\ell^{(\gamma)}$ (1), constitutes a Markov chain in a compact space. This chain admits at least one invariant distribution $\nu$ for which the following holds

$$\int g(\text{BN}_{0,\mathbf{1}_d}(H + \gamma W H))\mu(dW)\nu(dH) = \int g(H)\nu(dH), \tag{38}$$

for every bounded Borel function $g : \mathbb{R}^{d \times d} \to \mathbb{R}^d$. The above invariance property provides an interesting characterization of the invariant measure $\nu$. Particularly, we show in Lemma 13 that

$$\int r(H)\nu(dH) = \Omega(\sqrt{d}) \tag{39}$$

holds, where $r(H)$ is the established lower-bound on the rank (see Lemma 1). Under weak assumptions, the chain obey Birkhoff's Ergodicity, which yields that the average behaviour of the hidden representations is determined by the invariant measure $\nu$:

$$\lim_{L \to \infty} \frac{1}{L} \sum_{i=\ell} r(H_\ell^{(\gamma)}) = \int r(H)\nu(dH) \stackrel{(39)}{=} \Omega(\sqrt{d}). \tag{40}$$

Finally, the established lower bound in Lemma 1 allows us to directly extend this result to a lower bound on the soft rank itself. $\qquad\square$

**Characterizing the change in Frobenius norm** Recall the established lower bound on the rank denoted by $r(H)$, for which

$$r(H_\ell) = \frac{\text{Tr}(M_\ell)^2}{\|M_\ell\|_F^2} = \frac{d^2}{\|M_\ell\|_F^2} \tag{41}$$

holds for all $H_\ell$ defined in Eq. 1.[11] Therefore, $\|M_\ell\|_F^2$ directly influences $\text{rank}_\tau(H_\ell)$ (and also $\text{rank}(H_\ell)$) according to Lemma 1. Here, we characterize the change in $\|M(H)\|_F^2$ after applying one step of the recurrence in Eq. 7 to $H$, i.e. passing it trough one hidden layer. This yields

$$H_+ = (\text{diag}(M(H_\gamma(W))))^{-1/2} H_\gamma(W), \quad H_\gamma(W) = (I + \gamma W)H. \tag{42}$$

Let $M = M(H)$ and $M_+ = M(H_+)$ for simplicity. The next lemma estimates the expectation (taken over the randomness of $W$) of the difference between the Frobenius norms of $M_+$ and $M$.

**Lemma 7.** *If $W \sim \mu$ (defined in Def. 1), then*

$$\left(\mathbb{E}_W\|M_+\|_F^2 - \|M\|_F^2\right)/(\gamma^2) = \underbrace{2d^2 - 2\|M\|_F^2 - 8Tr(M^3) + 8Tr(diag(M^2)^2)}_{\delta_F(M)} + O(\gamma) \tag{43}$$

*holds as long as the support of distribution $\mathcal{P}$ (in Def. 1) lies in a finite interval $[-B, B]$.*

The proof of the above lemma is based on a Taylor expansion of the BN non-linear operator. We postpone the detailed proof to the end of this section. While the above equation seems complicated at first glance, it provides some interesting insights.

**Interlude: Intuition behind Lemma 7.** In order to gain more understanding of the implications of the result derived in Lemma 7, we make the simplifying assumption that all the rows of matrix $M$ have the same norm. We emphasize that this assumption is purely for intuition purposes and is not necessary for the proof of our main theorem. Under such an assumption, the next proposition shows that the change in the Frobenius norm directly relates to the spectral properties of matrix $M$.

**Proposition 8.** *Suppose that all the rows of matrix $M$ have the same norm. Let $\lambda \in \mathbb{R}^d$ contain the eigenvalues of matrix $M$. Then,*

$$Tr(M^3) = \|\lambda\|_3^3, \quad Tr(diag(M^2))^2 = \|\lambda\|^4/d, \quad \|M\|_F^2 = \|\lambda\|_2^2 \tag{44}$$

*holds and hence*

$$\delta_F(M) = \delta_F(\lambda) := 2d^2 - 2\|\lambda\|_2^2 - 8\|\lambda\|_3^3 + 8\|\lambda\|^4/d. \tag{45}$$

We postpone the proof to the end of this section. This proposition re-expresses the polynomial of Lemma 7 in terms of the eigenspectrum of $M$.

Based on the above proposition, we can make sense of interesting empirical observation reported in Figure 3.b. This figure plots the evolution of the eigenvalues of $M(H_\ell^{(\gamma)})$ after starting from a matrix $M(H_0)$ whose leading eigenvalue is large and all other eigenvalues are very small. We observe that a certain fraction of the small eigenvalues of $M(H_\ell^{(\gamma)})$ grow quickly with $\ell$, while the leading eigenvalue is decreases in magnitude. In the next example, we show that the result of the last proposition actually predicts this observation.

**Example 9.** *Suppose that $M$ is a matrix whose rows have the same norm. Let $\lambda_1 \geq \lambda_2, \ldots, \lambda_d$ be the eigenvalues associated with the matrix $M$ such that $\lambda_d = \lambda_{d-1} = \lambda_2 = \gamma^2$ and $\lambda_1 = d - \gamma^2(d-1)$. In this setting, Prop. 8 implies that $\mathbb{E}_W \|M_+\|_F^2 < \|M\|_F^2 - \gamma^4 d^2$ for a sufficiently small $\gamma$. This change has two consequences in expectation:(i.) the leading eigenvalue of $M_+$ is $O(-\gamma^4 d)$ smaller than the leading eigenvalue of $M$, and (ii.) some small eigenvalues of $M_+$ are greater than those of $M$ (see Fig. 3.b).*

We provide a more detailed justification for the above statement at the end of this section. This example illustrates that the change in Frobenius norm (characterized in Lemma 7) can predict the change in the eigenvalues of $M(H_\ell^{(\gamma)})$ (singular values of $H_\ell^{(\gamma)}$) and hence the desired rank. Inspired by this, we base the proof of Theorem 2 on leveraging the invariance property of the unique invariant distribution with respect to Frobenius norm – i.e. setting $g(H) = \|M(H)\|_F^2$ in Def. 2.

**An observation: regularity of the invariant distribution** We now return to the result derived in Lemma 7 that characterizes the change in Frobenius norm of $M(H)$ after the recurrence of Eq. (7). We show how such a result can be used to leverage the invariance property with respect to the Frobenius norm. First, we observe that the term $Tr(M(H)^3)$ in the expansion can be shown to dominate the term $Tr(diag(M(H)^2)^2)$ in expectation. The next definition states this dominance formally.

**Definition 3.** *(Regularity constant $\alpha$) Let $\nu$ be a distribution over $H \in \mathcal{H}$. Then the regularity constant associated with $\nu$ is defined as the following ratio:*

$$\alpha = \mathbb{E}_{H\sim\nu}\left[Tr\left(diag(M(H)^2)^2\right)\right] / \left(\mathbb{E}_{H\sim\nu}\left[Tr\left(M(H)^3\right)\right]\right). \tag{46}$$

The next lemma states that the regularity constant $\alpha$ associated with the invariant distribution $\nu_\gamma$ is always less than one. Our analysis will in fact directly rely on $\alpha < 1$.

**Lemma 10.** *Suppose that the chain $\{H_\ell^{(\gamma)}\}$ admits the unique invariant distribution $\nu_\gamma$ (in Def. 2). Then, the regularity constant of $\nu_\gamma$ (in Def. 3) is less than one for a sufficiently small $\gamma$.*

*Proof.* We use a proof by contradiction where we suppose that the regularity constant of distribution $\nu_\gamma$ is greater than one. In this case, we prove that the distribution cannot be invariant with respect to the Frobenius norm.

If the regularity constant $\alpha$ is greater than one, then

$$\mathbb{E}_{H\sim\nu_\gamma}\left[-Tr(M(H)^3) + Tr(diag(M(H)^2)^2)\right] \geq 0 \tag{47}$$

holds. According to Theorem 5, the rank of $M(H)$ is at least 2. Since the sum of the eigenvalues is constant $d$, the leading eigenvalue is less than $d$. This leads to

$$\|M(H)\|_F^2 = \sum_i \lambda_i^2 \leq \max_i \lambda_i \left(\sum_j \lambda_j\right) \leq d \max_i \lambda_i < d^2.$$

Plugging the above inequality together with inequality 47 into the established bound in Lemma 7 yields

$$\mathbb{E}_{W,H\sim\nu_\gamma}\left[\|M(H_+)\|_F^2 - \|M(H)\|_F^2\right] > 0 \tag{48}$$

for a sufficiently small $\gamma$. Therefore, $\nu_\gamma$ does not obey the invariance property for $g(H) = \|M(H)\|_F^2$ in Def. 2. $\qquad\square$

We can experimentally estimate the regularity constant $\alpha$ using the Ergodicity of the chain. Assuming that the chain is Ergodic[12],

$$\lim_{L\to\infty}\frac{1}{L}\sum_{\ell=1}^{L}g(H_\ell^{(\gamma)}) = \mathbb{E}_{H\sim\nu_\gamma}[g(H)] \tag{49}$$

holds almost surely for every Borel bounded function $g : \mathcal{H} \to \mathbb{R}$. By setting $g_1(H) = \mathrm{Tr}(M(H)^3)$ and $g_2(H) = \mathrm{Tr}(\mathrm{diag}(M(H)^2)^2)$, we can estimate $\mathbb{E}_{H\sim\nu_\gamma}[g_i(H)]$ for $i = 1$, and 2. Given these estimates, $\alpha$ can be estimated. Our experiments in Fig. 8 show that the regularity constant of the invariant distribution $\nu_\gamma$ is less than 0.9 for $d > 10$.

$$\gamma = 1 \qquad\qquad\qquad \gamma = 0.1$$

Figure 8: Regularity constant of the invariant distribution. The vertical axis is the estimated regularity constant $\alpha$ and the horizontal axis is $d$. We use $L = 10^5$ (in Eq. (49)).

**Interlude: intuition behind the regularity** We highlight the regularity constant does by itself not yield the desired rank property in Theorem 2. This is illustrated in the next example that shows how the regularity constant relates to the spectral properties of $M(H)$.

**Example 11.** *Suppose that the support of distribution $\nu$ contains only matrices $H \in \mathcal{H}$ for which all rows of $M(H)$ have the same norm. If the regularity constant of $\nu$ is greater than or equal to one, then all non-zero eigenvalues of matrix $M(H)$ are equal.*

A detailed justification of the above statement is presented at the end of this section. This example shows that the regularity constant does not necessarily relate to the rank of $H$, but instead it is determined by how much non-zero eigenvalues are close to each other. We believe that a sufficient variation in non-zero eigenvalues of $M(H)$ imposes the regularity of the law of $H$ with a constant less than one (i.e. $\alpha < 1$ in Def. 3). The next example demonstrates this.

**Example 12.** *Suppose the support of distribution $\nu$ contains matrices $H \in \mathcal{H}$ for which all rows of $M(H)$ have the same norm. Let $\lambda \in \mathbb{R}^d$ contain sorted eigenvalues of $M(H)$. If $\lambda_1 = \Theta(d^\beta)$ and $\lambda_i = o(d^\beta)$ for $i > 1$ and $\beta < 1$,[13] then the regularity constant $\alpha$ associated with $\nu$ is less than 0.9 for sufficiently large $d$.*

We later provide further details about this example.

**Invariance consequence** The next lemma establishes a key result on the invariant distribution $\nu_\gamma$.

**Lemma 13.** *Suppose that the chain $\{H_\ell^{(\gamma)}\}$ (see Eq. 7) admits the invariant distribution $\nu_\gamma$ (see Def. 2). If the regularity constant associated with $\nu_\gamma$ is $\alpha < 1$ (defined in Def. 3), then*

$$\mathbb{E}_{H\sim\nu_\gamma}\left[\|M(H)\|_F^2\right] \le d^{3/2}/\sqrt{1-\alpha} \tag{50}$$

*holds for a sufficiently small $\gamma$.*

*Proof.* Leveraging invariance property in Def. 2,

$$\mathbb{E}_{W,H\sim\nu_\gamma}\left[\|M(H_+)\|_F^2 - \|M(H)\|_F^2\right] = 0 \tag{51}$$

holds where the expectation is taken with respect to the randomness of $W$ and $\nu_\gamma$.[14] Invoking the result of Lemma 7, we get

$$\mathbb{E}_{H\sim\nu_\gamma}\left[2d^2 - 2\|M(H)\|_F^2 - 8\mathrm{Tr}(M(H)^3) + 8\mathrm{Tr}(\mathrm{diag}(M(H)^2)^2)\right] + \mathrm{O}(\gamma) = 0. \tag{52}$$

Having a regularity constant less than one for $\nu_\gamma$ implies

$$0 \le 2d^2 - \mathbb{E}_{H\sim\nu_\gamma}\left[2\|M(H)\|_F^2 - 8(1-\alpha)\mathrm{Tr}(M(H)^3)\right] \tag{53}$$

holds for sufficiently small $\gamma$. Let $\lambda \in \mathbb{R}^d$ be a random vector containing the eigenvalues of the random matrix $M(H)$.[15] The eigenvalues of $M^3$ are $\lambda^3$, hence the invariance result can be written alternatively as

$$0 \le 2d^2 - \mathbb{E}\left[2\|\lambda\|_2^2 - 8(1-\alpha)\|\lambda\|_3^3\right]. \tag{54}$$

The above equation leads to the following interesting spectral property:

$$\mathbb{E}\|\lambda\|_3^3 \le d^2/(1-\alpha). \tag{55}$$

A straightforward application of Cauchy-schwarz yields:

$$\|\lambda\|_2^2 = \sum_i \lambda_i^2 = \sum_i \lambda_i^{1/2}\lambda_i^{3/2} \le \sqrt{\sum_i \lambda_i \sum_j \lambda_i^3} \le \sqrt{d\|\lambda\|_3^3} \tag{56}$$

Given (i) the above bound, (ii) an application of Jensen's inequality, (iii) and the result of Eq. (55), we conclude with the desired result:

$$\mathbb{E}_{H\sim\nu_\gamma}\left[M(H)\right] = \mathbb{E}\left[\|\lambda\|_2^2\right] \overset{(i)}{\le} \mathbb{E}\sqrt{d\|\lambda\|_3^3} \overset{(ii)}{\le} \sqrt{d\mathbb{E}\|\lambda\|_3^3} \overset{(iii)}{\le} d^{3/2}/\sqrt{1-\alpha} \tag{57}$$

$\square$

Notably, the invariant distribution is observed to have a regularity constant less than 0.9 (in Fig. 8) for sufficiently large $d$. This implies that an upper-bound $\mathrm{O}\left(d^{3/2}\right)$ is achievable on the Frobenius norm. Leveraging Ergodicity (with respect to Frobenius norm in Eq. (49)), we experimentally validate the result of the last lemma in Fig. 9.

$$\gamma = 1 \qquad\qquad\qquad \gamma = 0.1$$

Figure 9: Dependency of $\mathbb{E}_{\nu_\gamma}\|M(H)\|_F^2$ on $d$. The horizontal axis is $\log_2(d)$ and the vertical axis shows $\log_2(\frac{1}{L}\sum_{\ell=1}^L \|M(H_\ell^{(\gamma)})\|_F^2)$ for $L = 10^5$. The green dashed-line plots $\log_2(d^{1.5})$.

**Proof of the Main Theorem** Here, we give a formal statement of the main Theorem that contains all required additional details (which we omitted for simplicity in the original statement).

**Theorem 14** (Formal statement of Theorem 2). *Suppose that rank$(X) = d$, $\gamma$ is sufficiently small, and all elements of the weight matrices $\{W_\ell\}$ are drawn i.i.d. from a zero-mean, unit variance distribution whose support lies in $[-B, B]$ and its law is symmetric around zero. Furthermore, assume that the Markov chain $\{H_\ell^{(\gamma)}\}$ (defined in Eq. 1) admits a unique invariant distribution. Then, the regularity constant $\alpha > 0$ associated with $\nu_\gamma$ (see Def. 3) is less than one and the following limits exist such that*

$$\lim_{L\to\infty} \frac{1}{L} \sum_{\ell=1}^{L} rank_\tau(H_\ell^{(\gamma)}) \geq \lim_{L\to\infty} \frac{(1-\tau)^2}{L} \sum_{\ell=1}^{L} r(H_\ell^{(\gamma)}) \geq (1-\tau)^2 (1-\alpha)^{1/2}\sqrt{d} \quad (58)$$

*holds almost surely for all $\tau \in [0, 1]$. Assuming that the regularity constant $\alpha$ does not increase with respect to $d$, the above lower-bound is proportional to $(1-\alpha)^{1/2}\sqrt{d} = \Omega(\sqrt{d})$.*

Remarkably, we experimentally observed (in Fig. 8) that the regularity constant $\alpha$ is decreasing with respect to $d$. Examples 11 and 12 provide insights about the regularity constant. We believe that it is possible to prove that the constant $\alpha$ is non-increasing with respect to $d$.

*Proof of Theorem 2 .* Lemma 10 proves that the regularity constant $\alpha$ is less than one for the unique invariant distribution. Suppose that $H \in \mathcal{H}$ is a random matrix whose law is the one of the unique invariant distribution of the chain. For $H \in \mathcal{H}$, we get $\mathrm{Tr}(M(H)) = d$. A straightforward application of Jensen's inequality yields the following lower bound on the expectation of $r(H)$ (i.e. the lower bound on the rank):

$$\mathbb{E}\left[r(H)\right] = \mathbb{E}\left[\mathrm{Tr}(M(H))^2/\|M(H)\|_F^2\right] = \mathbb{E}\left[d^2/\|M(H)\|_F^2\right] \geq d^2/\mathbb{E}\left[\|M(H)\|_F^2\right] \quad (59)$$

where the expectation is taken over the randomness of $H$ (i.e. the invariant distribution). Invoking the result of Lemma 13, we get an upper-bound on the expectation of the Frobenius norm – in the right-side of the above equation. Therefore,

$$\mathbb{E}\left[r(H)\right] \geq \sqrt{(1-\alpha)d} \quad (60)$$

holds. The uniqueness of the invariant distribution allows us to invoke Birkhoff's Ergodic Theorem for Markov Chains (Theorem 5.2.1 and 5.2.6 [11]) to get

$$\lim_{L\to\infty} \frac{1}{L} \sum_{\ell=1}^{L} r(H_\ell^{(\gamma)}) = \mathbb{E}\left[r(H)\right] \geq \sqrt{(1-\alpha)d}. \quad (61)$$

The established lower bound on $rank_\tau(H_\ell^{(\gamma)})$ –in terms of $r(H_\ell^{(\gamma)})$– in Lemma 1 concludes

$$\lim_{L\to\infty} \frac{1}{L} \sum_{\ell=1}^{L} rank_\tau(H_\ell^{(\gamma)}) \geq \lim_{L\to\infty} \frac{(1-\tau)^2}{L} \sum_{\ell=1}^{L} r(H_\ell^{(\gamma)}) \geq (1-\tau)^2 \sqrt{(1-\alpha)d}. \quad (62)$$

$\square$

As shown in the following corollary, one can extend the result of Theorem 14 for any finite $\ell$.

**Corollary 15.** *Under the setting of Thm. 14, rank$(H_\ell) = \Omega(\sqrt{d})$ holds almost surely for all finite integer $\ell$. Assuming that $\{rank_\tau(H_\ell)\}$ is a monotonically no-increasing sequence, then $rank_\tau(H_\ell) = \Omega((1-\tau)^2\sqrt{d})$ holds almost surely for all finite $\ell$.*

*Proof.* The proof is based on the no-increasing property of the rank[16]. Next lemma presents a straightforward implication of this property.

**Lemma 16.** *Consider a sequence of non-increasing bounded finite integers $\{y_k\}_{k=1}^{\infty}$. If $\lim_{N\to\infty} \sum_{k=1}^{N} y_k/N$ exists and is greater than $\alpha$, then $y_k \geq \alpha$ for all finite $k$.*

The proof of the last lemma is provided at the end of this section. Replacing the result of Thm. 14 into the above lemma concludes the proof of the corollary.

$\square$

**A remark on the number of hidden units.**    The focus of our analysis was networks with the same number of hidden units in each layer. Yet, this result extends to more general architectures. Most of modern neural architectures consists of blocks in which the number of hidden units are constant. For example, VGG19-Nets and ResNets are consist of blocks convolutional layers with 64, 128, 256, and 512 channels where the number channels are equal in each block. An analogy of such an architecture is an MLP with different blocks of hidden layers where the numbers of hidden units are the same in each block. According to Cor. 15, the rank preservation property holds in each block after applying BN. In this way, one can extend the established results of Thm. 14 and Cor. 15 to a general family of architectures with varying number of hidden units.

**Postponed proofs.**

*Proof of Lemma 7.* The proof is based on a Taylor expansion of the BN non-linear recurrence function, which we restate here for simplicity:

$$H_+ = (\text{diag}(M(H_\gamma)))^{-1/2} H_\gamma, \quad H_\gamma = (I + \gamma W)H \tag{63}$$

Consider the covariance matrices $M = M(H)$ and $M_+ = M(H_+)$ which obey

$$M_\gamma := M(H_\gamma) = M + \Delta M, \quad \Delta M := \gamma W M + \gamma M W^\top + \gamma^2 W M W^\top \tag{64}$$

$$[M_+]_{ij}^2 = g_{ij}(M_\gamma) = [M_\gamma]_{ij}^2 / [M_\gamma]_{ii} [M_\gamma]_{jj} \tag{65}$$

For the sake of simplicity, we use the compact notation $g := g_{ij}$ for $i \neq j$. We further introduce the set of indices $S = \{ii, ij, jj\}$. A taylor expansion of $g$ at $M$ yields

$$\mathbb{E}_W [g(M_\gamma)] = g(M) + \underbrace{\sum_{pq \in S} \left( \frac{\partial g(M)}{\partial M_{pq}} \right) \mathbb{E}_W [\Delta M_{pq}]}_{T_1}$$

$$+ \underbrace{\frac{1}{2} \sum_{pq, km \in S} \left( \frac{\partial^2 g(M)}{\partial M_{pq} \partial M_{km}} \right) \mathbb{E}_W [\Delta M_{pq} \Delta M_{km}]}_{T_2} + \text{O}(\gamma^3). \tag{66}$$

Note that the choice of the element-wise uniform distribution over $[-\sqrt{3}, \sqrt{3}]$ allows us to deterministically bound the Taylor remainder term by $\text{O}(\gamma^3)$. Now, we compute the derivatives and expectations that appear in the above expansion individually. Let us start with the term $T_1$. The first-order partial derivative term in $T_1$ is computed bellow.

$$\frac{\partial g(M)}{\partial M_{pq}} = \begin{cases} -M_{ij}^2 / (M_{ii}^2 M_{jj}) = -g(M) & pq = \{ii, jj\} \\ 2M_{ij} / (M_{ii} M_{jj}) & pq = \{ij\}. \end{cases} \tag{67}$$

The expectation term in $T_1$ is

$$\mathbb{E}_W [\Delta M_{pq}] = \begin{cases} 0 & pq = \{ij\} \\ \gamma^2 \sum_{k=1}^d M_{kk} = \gamma^2 d & pq = \{ii, jj\}. \end{cases} \tag{68}$$

Given the above formula, we reach the following compact expression for $T_1$:

$$T_1 = -2\gamma^2 d g(M). \tag{69}$$

The compute $T_2$ we need to compute second-order partial derivatives of $g$ and also estimate the following expectation:

$$\mathbb{E}_W [\Delta M_{pq} \Delta M_{km}] = \gamma^2 \left( \underbrace{\mathbb{E}_W \left[ [WM + MW^\top]_{pq} [WM + MW^\top]_{km} \right]}_{K_{pq, km}} \right) + \text{O}(\gamma^3). \tag{70}$$

We now compute $K_{pq, km}$ in the above formula

$$K_{\alpha, \beta} = \begin{cases} \sum_k M_{kj}^2 + \sum_n M_{in}^2 & \alpha = \{ij\}, \beta = \{ij\} \\ 2 \sum_k M_{kj} M_{ki} & \alpha = \{ij\}, \beta = \{ii\} \\ 4 \sum_k M_{ki}^2 & \alpha = \{ii\}, \beta = \{ii\} \\ 0 & \alpha = \{ii\}, \beta = \{jj\} \end{cases} \tag{71}$$

The second-order partial derivatives of $g$ reads as

$$\frac{\partial^2 g(M)}{\partial M_\alpha \partial M_\beta} = \begin{cases} 2 & \alpha = \{ij\}, \beta = \{ij\} \\ -2M_{ij} & \alpha = \{ij\}, \beta = \{ii\} \\ +2M_{ij}^2 & \alpha = \{ii\}, \beta = \{ii\} \\ M_{ij}^2 & \alpha = \{jj\}, \beta = \{ii\} \end{cases} \tag{72}$$

Now, we replace the computed partial derivatives and the expectations into $T_2$:

$$T_2 = \sum_k M_{kj}^2 + \sum_n M_{in}^2 - 8 \sum_k M_{kj} M_{ij} M_{ki} + 4 \sum_k M_{ij}^2 M_{ki}^2 + 4 \sum_k M_{ij}^2 M_{kj}^2 \qquad (73)$$

Plugging terms $T_1$ and $T_2$ into the Taylor expansion yields

$$\mathbb{E}_W \left[ g_{ij}(M_+) - g_{ij}(M) \right] / (\gamma^2)$$
$$= \sum_k M_{kj}^2 + \sum_n M_{in}^2 - 2 d g_{ij}(M) - 8 \sum_k M_{kj} M_{ij} M_{ki} + 4 \sum_k M_{ij}^2 M_{ki}^2 + 4 \sum_k M_{ij}^2 M_{kj}^2 + \mathrm{O}(\gamma) \qquad (74)$$

Summing over $i \neq j$ concludes the proof (note that the diagonal elements are one for the both of matrices $M$ and $M_+$). $\qquad \square$

*Proof of Proposition 8.* Consider the spectral decomposition of matrix $M$ as $M = U\mathrm{diag}(\lambda)U^\top$, then $M^k = U\mathrm{diag}(\lambda^k)U^\top$. Since $\mathrm{Tr}(M^k)$ is equal to the sum of the eigenvalues of $M^k$, we get

$$\mathrm{Tr}(M^k) = \sum_{i=1}^d \lambda_i^k = \|\lambda\|_k^k \qquad (75)$$

for $k = 2$ and $k = 3$. The sum of the squared norm of the rows in $M$ is equal to the Frobenius norm of $M$. Assuming that the rows have equal norm, we get

$$\sum_{k=1}^d M_{ik}^2 = \sum_{i=1}^d \sum_{k=1}^d M_{ik}^2 / d = \|M\|_F^2 / d = \|\lambda\|_2^2 / d. \qquad (76)$$

Therefore,

$$\mathrm{Tr}(\mathrm{diag}(M^2)^2) = \sum_{i=1}^d \left( \sum_{k=1}^d M_{ik}^2 \right)^2 = \|\lambda\|_2^4 / d \qquad (77)$$

holds.

$\qquad \square$

*Details of Example 9.* Under the assumptions stated in Example 9, we get

$$\|\lambda\|_2^2 \approx d^2 - 2\gamma^2 d, \quad \|\lambda\|_3^3 \approx d^3 - 3\gamma^2 d^2, \quad \|\lambda\|_2^4 \approx d^4 - 4\gamma^2 d^3 \qquad (78)$$

where the approximations are obtained by a first-order Taylor approximation of the norms at $\lambda' = (d, 0, \ldots, 0)$, and all small terms $o(\gamma^2)$ are omitted. Using the result of Proposition 8, we get

$$\mathbb{E}\left[ \|M_+\|_F^2 \right] - \mathbb{E}\left[ \|M\|_F^2 \right] \approx \gamma^2 \delta_F(\lambda) \approx \mathrm{O}(-\gamma^4 d^2). \qquad (79)$$

Let $\lambda_+$ be the eigenvalues of the matrix $M_+$, then

$$\sum_{i=1}^d \mathbb{E}[\lambda_+^2]_i - \lambda_i^2 = \mathrm{O}(-\gamma^4 d^2) \qquad (80)$$

$$\implies \max_i \mathbb{E}[\lambda_+^2]_i - \lambda_1^2 \leq \mathrm{O}(-\gamma^4 d^2) + \sum_{i=2}^d \lambda_i^2 \leq \mathrm{O}(-\gamma^4 d^2) + \gamma^4 d = \mathrm{O}(-\gamma^4 d^2). \qquad (81)$$

Let $j = \arg\max_i \mathbb{E}\left[ [\lambda_+]_i^2 \right]$. A straight-forward application of Jensen's inequality yields

$$\mathbb{E}\left[ [\lambda_+]_j \right] \leq \sqrt{\mathbb{E}\left[ [\lambda_+]_j^2 \right]} \leq \lambda_1 - \mathrm{O}(\gamma^4 d). \qquad (82)$$

Hence the leading eigenvalue of $M_+$ is smaller than the one of $M$. Since the sum of eigenvalues $\lambda_+$ and $\lambda$ are equal, some of the eigenvalues $\lambda_+$ are greater than those of $\lambda$ (in expectation) to compensate $\mathbb{E}[\lambda_+]_j < \lambda_1$. $\qquad \square$

*Details of Example 11.* Invoking Prop. 8, we get

$$\mathbb{E}\left[\text{Tr}(M(H)^3)\right] = \|\lambda\|^3, \quad \mathbb{E}\left[\text{diag}(M(H)^2)^2\right] = \|\lambda\|_2^4/d, \tag{83}$$

where $\lambda \in \mathbb{R}^d$ contains the eigenvalues of $M(H)$. Since $H \in \mathcal{H}$, $\|\lambda\|_1 = d$. If the regularity constant is greater than or equal to one, then

$$\|\lambda\|_3^3 \leq \|\lambda\|_2^4/d = \|\lambda\|_2^4/\|\lambda\|_1. \tag{84}$$

A straightforward application of Cauchy-Schwartz yields:

$$\|\lambda\|_2^4 = \sum_{i=1}^d \sum_{j=1}^d \lambda_i^2 \lambda_j^2 = \sum_{i=1}^d \sum_{j=1}^d (\lambda_i \lambda_j)^{1/2} (\lambda_i \lambda_j)^{3/2}$$

$$\leq \sqrt{\left(\sum_{i,j} \lambda_i \lambda_j\right)\left(\sum_{i,j} \lambda_i^3 \lambda_j^3\right)} = \|\lambda\|_1 \|\lambda\|_3^3 \tag{85}$$

The above result together with inequality 84 yields that

$$\|\lambda\|_3^3 = \|\lambda\|_2^4/d = \|\lambda\|_2^4/\|\lambda\|_1. \tag{86}$$

Finally, the above equality is met only when all non-zero eigenvalues are equal.

$\square$

*Details of Example 12.* Since $\lambda_1 = \Theta(d^\beta)$ and $\lambda_{i>1} = o(d^\beta)$, we get

$$\|\lambda\|_3^3 = \Theta(d^{3\beta}), \quad \|\lambda\|_2^2 = \Theta(d^{2\beta}). \tag{87}$$

Thus, Prop. 8 yields

$$\mathbb{E}\left[\text{Tr}(M^3)\right] = \Theta(d^{3\beta}), \quad \mathbb{E}\left[\text{Tr}(\text{diag}(M^2)^2)\right] = \|\lambda\|_2^4/d = \Theta(d^{4\beta-1}) \tag{88}$$

Therefore,

$$\alpha = \lim_{d\to\infty} \frac{\mathbb{E}\left[\text{Tr}(\text{diag}(M^2)^2)\right]}{\mathbb{E}\left[\text{Tr}(M^3)\right]} = \text{O}(d^{\beta-1}) = 0. \tag{89}$$

As a result, $\alpha$ is less than 0.9 for sufficiently large $d$. $\square$

*Proof of Lemma 16.* The proof is based on a contradiction. Suppose that there exits a finite $n$ such that $y_n < \alpha$. Since the sequence is non-increasing, $y_m < \alpha$ for holds for all $m > n$. This yields

$$\lim_{N\to\infty} \sum_{k=1}^N y_k/N = \lim_{N\to\infty} \left(\sum_{k>n}^N y_k/N + \sum_{k\leq n} y_k/N\right) \tag{90}$$

$$< \frac{(N-n)}{N}\alpha + \lim_{N\to\infty} \sum_{k\leq n} y_k/N \tag{91}$$

$$= \frac{(N-n)}{N}\alpha, \tag{92}$$

where we used the fact that all $y_k$ are bounded. The above result contradicts the fact that $\lim_{n\to\infty} \sum_{k=1}^N y_k/N > \alpha$. $\square$

## F  Analysis for Vanilla Linear Networks.

In this section, we prove Lemma 3 that states the rank vanishing problem for vanilla linear networks. Since the proof relies on existing results on products of random matrices (PRM) [9], we first shortly review these results. Let $T$ be the set of $d \times d$ matrices. Then, we review two notions for $T$: contractiveness and strong irreducibility.

**Definition 4** (Contracting set [9]). *$T$ is contracting if there exists a sequence $\{M_n \in T, n \geq 0\}$ such that $M_n/\|M_n\|$ converges to a rank one matrix.*

**Definition 5** (Invariant union of proper subspaces [9]). *Consider a family of finite proper linear subspace $V_1, \ldots, V_k \subset \mathbb{R}^d$. The union of these subspaces is invariant with respect to $T$, if $Mv \in V_1$ or $V_2$ or $\ldots$ or $V_k$ holds for $\forall v \in V_1$ or $V_2$ or $\ldots$ or $V_k$ and $\forall M \in T$.*

**Example 17.** *Consider the following sets*

$$
T = \left( \begin{bmatrix} 0 & 1 \\ 1 & 0 \end{bmatrix} \right), \quad V_1 = \left( span(\underbrace{[0,1]}_{v_1}) \right), \quad V_2 = \left( span(\underbrace{[1,0]}_{v_2}) \right);
$$

*then, union of $V_1$ and $V_2$ is invariant with respect to $T$ because $\alpha T v_1 \in V_2$ and $\alpha T v_2 \in V_1$ hold for $\alpha \neq 0$.*

**Definition 6** (Strongly irreducible set [9]). *The set $T$ is strongly irreducible if there does not exist a finite family of proper linear subspaces of $\mathbb{R}^d$ such that their union is invariant with respect to $T$.*

For example, the set $T$ defined in Example 17 is not strongly irreducible.

**Lemma 18** (Thm 3.1 of [9]). *Let $W_1, W_2, \ldots$ be random $d \times d$ matrices drawn independently from a distribution $\mu$. Let $B_n = \prod_{k=1}^{n} W_k$. If the support of $\mu$ is strongly irreducible and contracting, then any limit point of $\{B_n/\|B_n\|\}_{n=1}^{\infty}$ is a rank one matrix almost surely.*

This result allows us to prove Lemma 3.

*Proof of Lemma 3.* Recall the structure of the random weight matrices as $\widehat{W}_k = I + \gamma W_k$ where the coordinates $W_k$ are i.i.d. from (a.) standard Gaussian, (b.) uniform$[-\sqrt{3}, \sqrt{3}]$ (i.e. with variance 1). One can readily check that for the Gaussian weights, the contracting and strong irreducibility hold and one can directly invoke the result of lemma 18 to get part (a.) of Lemma 3. Now, we prove part (b.). Let $m$ be a random integer that obeys the law $p(m = k) = 2^{-k}$. Given the random variable $m$, we define the random matrix $Y = \prod_{k=1}^{m} \widehat{W}_k$ and use the notation $\mu'$ for its law. Let $\{Y_i = \prod_{j=1}^{m_i} \widehat{W}_k\}_{i=1}^{k}$ be drawn i.i.d. from $\mu'$. Then, $C_k := Y_k \ldots Y_2 Y_1$ is distributed as $B_{\ell_k} := \widehat{W}_{\ell_k} \ldots \widehat{W}_2 \widehat{W}_1$ for $\ell_k = \sum_{i=1}^{k} m_i$. We prove that every limit point of $\{C_k/\|C_k\|\}$ converges to a rank one matrix, which equates the convergence of limit points of $\{B_{\ell_k}/\|B_{\ell_k}\|\}$ to a rank one matrix. To this end, we prove that the support of $\mu'$ denoted by $T_{\mu'}$ is contractive and strongly contractive. Then, Lemma 18 implies that the limit points of $\{C_k/\|C_k\|\}$ are rank one.

**Contracting.** Let $e_1 \in \mathbb{R}^d$ be the first standard basis vector. Since $A_n := (I + \gamma e_1 e_1^\top)^n \in T_{\mu'}$ and its limit point $\{A_n/\|A_n\|\}$ converges to a rank one matrix, $T_{\mu'}$ is contractive.

**Strong irreduciblity.** Consider an arbitrary family of linear proper subspace of $\mathbb{R}^d$ as $\{V_1, \ldots, V_q\}$. Let $v$ be an arbitrary unit norm vector which belongs to one of the subspaces $\{V_i\}_{i=1}^{q}$. Given $v$, we define an indexed family of matrices $\{M_\alpha \in T_{\mu'} | \alpha \in \mathbb{R}^d, |\alpha_i| \leq 1\}$ such that

$$
M_\alpha = I + \frac{\gamma}{d} \sum_{i=1}^{d} \alpha_i e_i v^\top \in T_{\mu'}, \tag{93}
$$

where $e_i$ is the i-th standard basis[17]. Then, we get

$$
M_\alpha v = v + \frac{\gamma}{d} \sum_{i=1}^{d} \alpha_i e_i. \tag{94}
$$

Therefore, $\{M_\alpha v \||\alpha_i| \leq 1\}$ is not contained in any union of finite proper $(m < k)$-dimensional linear subspace of $\mathbb{R}^d$, hence $T_{\mu'}$ is strongly irreducible.

$\square$

# G    Details: Pretraining algorithm

In Section 4, we introduced a pre-training method that effectively obtains a better optimization performance compared ot BN. In this section, we provide more details about the pre-training step. Recall $X \in \mathbb{R}^{d \times N}$ is a minibatch of $d$-dimensional inputs of size $N$. Let $H_L(X) \in \mathbb{R}^{d \times N}$ be the hidden representation of input $X$ in the last layer of a MLP. Using gradient descent method, we optimize $r(H_L(X))$ –with respect to the parameters of networks– over different minibatches $X$. Algorithm 1 presents our pretraining method. As can be seen, the procedure is very simple.

---

**Algorithm 1** Pretraining

---

1: **Input:** Training set $S$, a network with parameters $\Theta$ and $L$ layers, and constant $N, M$, and $T$
2: **for** $k = 1, 2, \ldots, M$ **do**
3:    Draw minibatch $X_k$ of size $N$ i.i.d. from $S$
4:    **for** $t = 1, 2, \ldots, T$ **do**
5:       Take one GD step on $r(H_L(X_k))$ w.r.t $\Theta$.
6:    **end for**
7: **end for**
8: **return** $\Theta$.

---

# H    Details: Why the rank matters for gradient based learning.

We now provide an intuitive explanation of why rank one hidden representations prevent randomly initialized networks from learning. Particularly, we argue that these networks essentially map all inputs to a very small subspace[18] such that the final classification layer can no longer disentangle the hidden representations. As a result, the gradients of that layer also align, yielding a learning signal that becomes *independent* of the input.

To make this claim more precise, consider training the linear network from Eq. (6) on a dataset $X \in \mathbb{R}^{d \times N}$, where $x_i \in \mathbb{R}^d$ with $d_{out}$ targets $y_i \in \mathbb{R}^{d_{out}}, i = 1, \ldots, N$. Each column $\widehat{H}_{L,i}^{(\gamma)}$ of the hidden representations in the last hidden layer $\widehat{H}_L^{(\gamma)}$ is the latent representation of datapoint $i$, which is fed into a final classification layer parametrized by $W_{L+1} \in \mathbb{R}^{d_{out} \times d}$. We optimize $\mathcal{L}(\mathbf{W})$, where $\mathbf{W}$ is a tensor containing all weights $W_1, \ldots, W_{L+1}$ and $\widehat{H}_{L,i}^{(\gamma)}$ is a function of $W_1, \ldots, W_L$ (as detailed in Eq. (6):

$$\min_{\mathbf{W}} \mathcal{L}(\mathbf{W}) = \sum_{i=1}^{N} \underbrace{\ell \left( y_i, W_{L+1} \widehat{H}_{L,i}^{(\gamma)} (W_1, ..., W_L) \right)}_{:= \mathcal{L}_i(\mathbf{W})}, \tag{95}$$

and $\ell : \mathbb{R}^{d_{out}} \to \mathbb{R}^+$ is a differentiable loss function. Now, if the the hidden representations become rank one (as predicted by Lemma 3 and Fig. 2), one can readily check that the stochastic gradients of any neuron $k$ in the last linear layer, i.e., $\nabla_{W_{L,[k,:]}} \mathcal{L}_i(\mathbf{W}) = (\nabla \ell_i)_k \widehat{H}_{L,i}^{(\gamma)}$, align for both linear and ReLU networks.

**Proposition 19.** *Consider a network with rank one hidden representations in the last layer* $\widehat{H}_L^{(\gamma)}(W_1, ..., W_L)$, *then for any neuron $k$ and any two datapoints $i, j$ with non-zero errors $\mathcal{L}_i$ and $\mathcal{L}_j$ we have*

$$\nabla_{W_{L+1,[k,:]}} \mathcal{L}_i(\mathbf{W}) = \underbrace{\frac{c(\nabla \ell_i)_k}{(\nabla \ell_j)_k}}_{\in \mathbb{R}} \nabla_{W_{L+1,[k,:]}} \mathcal{L}_j(\mathbf{W}) \tag{96}$$

$\forall i, j$. *That is, all stochastic gradients of neuron $k$ in the final classification layer align along one single direction in $\mathbb{R}^d$.*

*Proof.* The result follows directly from a simple application of the chain rule

$$\frac{\partial \mathcal{L}_i(\mathbf{W})}{\partial W_{L+1}} = \frac{\partial \ell(\mathbf{y}_i, W_{L+1}\widehat{H}_{L,i}^{(\gamma)})}{\partial W_{L+1}\widehat{H}_{L,i}^{(\gamma)}} \frac{\partial W_{L+1}\mathbf{h}_{L,i}}{\partial W_{L+1}} = \nabla_{W_{L+1}\widehat{H}_{L,i}^{(\gamma)}}\ell(\mathbf{y}_i, W_{L+1}\mathbf{h}_{L,i})(\widehat{H}_{L,i}^{(\gamma)})^{\mathsf{T}}$$

$$= \begin{bmatrix} \nabla \ell_{i,1}\widehat{H}_{L,i,1}^{(\gamma)}, \dots, \nabla \ell_{i,1}\widehat{H}_{L,i,d}^{(\gamma)} \\ \ddots \\ \nabla \ell_{i,d_{out}}\widehat{H}_{L,i,1}^{(\gamma)}, \dots, \nabla \ell_{i,d_{out}}\widehat{H}_{L,i,d}^{(\gamma)} \end{bmatrix} \in \mathbb{R}^{d_{out}\times d} \tag{97}$$

The same holds for $j$. Now, if $\widehat{H}_{L,i}^{(\gamma)} = c\widehat{H}_{L,i}^{(\gamma)}, c \in \mathbb{R} \setminus \{0\}$ then

$$\left(\frac{\partial \mathcal{L}_i(\mathbf{W})}{\partial W_{L+1}}\right)_{k,:} = c\underbrace{\frac{\nabla \ell_{i,k}}{\nabla \ell_{j,k}}}_{\in \mathbb{R}}\left(\frac{\partial \mathcal{L}_j(\mathbf{W})}{\partial W_{L+1}}\right)_{k,:}$$

$\square$

To validate this claim, we again train CIFAR-10 on the VGG19 network from Figure 5 (top).

Figure 10: **Directional gradient vanishing** CIFAR-10 on a VGG19 network with BN, SGD, SGD with 100x learning rate and SGD on random data. Average and 95% confidence interval of 5 independent runs.

As expected, the network shows perfectly aligned gradients without BN (right hand side of Fig. 10), which renders it un-trainable. In a next step, we replace the input by images generated randomly from a uniform distribution between 0 and 255 and find that SGD takes almost the exact same path on this data (compare log accuracy on the left hand side). Thus, our results suggest that the commonly accepted vanishing gradient *norm* hypothesis is not descriptive enough since SGD does not take small steps into the *right* direction- but into a *random* one after initialization in deep neural networks. As a result, even a 100x increase in the learning rate does not allow training. We consider our observation as a potential starting point for novel theoretical analysis focusing on understanding the propagation of information through neural networks, whose importance has also been highlighted by [7].

# I  Additional Experiments

Figure 11: CIFAR-10: Same setting as Fig.4 but now showing accuracy and test loss

**Outperforming** BN  The following Figure shows the result of the experiment of Fig. 4 that is repeated for FashionMNIST dataset. As can be seen, overfitting tends to happen whenever a certain accuracy is achieved on the training set, regardless of the actual method that is used for optimization.

Figure 12: Results of Fig.4 for FashionMNIST

**Breaking** BN  In the following result, we repeated the experiment of Fig. 5 for ResNets.

Figure 13: **Breaking Batchnorm:** CIFAR-10 on a ResNet-50 with standard PyTorch initialization as well as a uniform initialization of same variance in $\mathbb{R}^+$. Average and 95% confidence interval of 5 independent runs. This plot also shows results for a BN network without mean deduction/adaption, validating our claim from Section 2.