[Reviews · NeurIPS 2020]

Review 1

Summary and Contributions: This paper studies the effect of adding batch normalization (BN) layers on the rank of the data matrix as it passes through the network. The authors prove theoretically that the rank of the data matrix when passed through a network with BN does not collapse to rank one, which is what occurs when the network does not have BN layers. The theoretical results are accompanied by experimental results which highlight the phenomenon in networks trained on and operating on real and simulated data.

Strengths: This paper is well motivated and grounded. While I did not completely check the proofs, the application of markov chain theory to analyzing representations learned by deep networks is an interesting approach and might be applicable beyond studying BN. The paper has an easy to follow narrative that highlights and explains the phenomenon. The authors also consider how initialization interacts with batch normalization and the rank of the representations.

Weaknesses: While there are not many technical or experimental weaknesses in this paper, I wonder whether rank preserving transformations are important in other learning models - say linear ones or kernel machines, etc. It could be the case that this is a phenomenon exclusive to deep networks and other models are not relevant. Another issue is that in the case of binary classification one could still perform the task when rank collapse happens, as long as the relevant discriminatory signal is captured by the principal direction that the data is collapsed to. I would like to know if the authors agree or disagree with this hypothetical. Finally I do not think the authors address the case where the networks are overparameterized and d>N in each layer. Then the controlling factor on the rank of the representations is N and not \Omega(\sqrt{d}). How do the authors think their results adapt to this situation.

Correctness: Yes the theory and experiments seem reasonable.

Clarity: Yes the paper is very well presented and was easy to read.

Relation to Prior Work: Yes the authors discuss other possible explanations for batch normalization and point out how their approach might fill in some gaps.

Reproducibility: Yes

Additional Feedback: Update after reading reviews and author response: The authors have addressed all my questions convincingly in their response. I hope they include their responses to these questions in the final version of the paper. I vote for acceptance.


Review 2

Summary and Contributions: The aim of the paper is to explore and evaluate the importance of batch normalization in deep learning networks initialized with random weights. This is an important point since most of the DNNs are trained with this method.

Strengths: - The paper provides a useful insight between batch-normalization and stability of the rank through the layers - Interestingly the authors prove that batch-normalization avoid rank collapse for linear NNn, while the importance of the rank in gradient based learning is experimentally evaluated. - The random initialization setting is widely used and relevant to the scientific community.

Weaknesses: - The proposed solution aspires at explaining an important behavior of DNNs. Unfortunately it does not provide a constructive approach to build new DNNs relying on / exploring batch-normalization. From this perspective the discussion paragraph is interesting even if further in-depth analysis would be required.

Correctness: The employed methods are correct

Clarity: The paper is generally well written.

Relation to Prior Work: The paper is well contextualized in the related literature.

Reproducibility: Yes

Additional Feedback: The overall impression of the paper is positive even if it is not fully clear how broad is the impact of what proposed.


Review 3

Summary and Contributions: This paper shows that BatchNorm can effectively prevent rank collapse in DNNs, both theoretically and empirically. The theoretical result holds for linear MLPs, but is also shown to be true empirically on a variety of commonly used neural nets trained on standard benchmark datasets. Rank collapse can cause gradients to become independent of the data, thus interfering with learning. In addition, the authors propose a novel pre-training method for SGD to avoid rank collapse in vanilla neural nets.

Strengths: -The theoretical claims are sound and the experiments are extensive. -The result is significant. There has been a line of work understanding batch normalization. This paper points out some insufficiencies of previous work, and develops new theories for the efficacy of BatchNorm. Moreover, the theoretical results are supported by experiments on standard deep neural nets. -The contribution is novel. The pre-training step for SGD is an interesting technique to use when BatchNorm isn’t the most efficient solution to avoid rank collapse.

Weaknesses: -The theoretical result seems to hold only when \gamma is small.

Correctness: Yes.

Clarity: Yes.

Relation to Prior Work: Yes.

Reproducibility: Yes

Additional Feedback: -In the warmup analysis, Lemma 6 in the appendix, \gamma is taken to be at most 1/8Bd. How does this result apply to the overparameterized regime, when d is large? Similarly, in Theorem 14, \gamma needs to be sufficiently small, meaning that the skip connection strength is high. Does this hold in practice? -In appendix I, figure 11, pre-trained SGD seems to overfit with 30 hidden layers. It seems that even though pre-trained SGD accelerates training, it might not generalize as well when the net is not very deep? -To what extent can pre-training replace BatchNorm? Are there experiments with pre-training in architectures like VGG?

[Author Response · NeurIPS 2020]

We sincerely thank all reviewers for their meaningful and detailed comments that will help improve our work. In the following, we address the raised concerns one by one:

**Reviewer 1.** We thank Reviewer 1 for his/her constructive feedback. (i) Linear- and kernel models: We consider rank collapse to be a phenomenon that is specific to the compositional structure of deep networks and hence shallow models are naturally robust against it. (ii) Binary classification: we agree that mapping all data points to a line does not impede this per se but we would like to point out that the direction of the rank collapse after initialization is in fact random and *independent* from the data, which makes it very unlikely to coincide with the principal direction in the data. Particularly so in very high dimensional neural networks. Fig. 1 shows that rank collapse remains an issue when reducing CIFAR10 to a binary classifcation task (using class 0 and 1 only). (iii) Networks wider than the input: Good point, we will make sure to clarify this in the paper. If $N < c\sqrt{d}$ for some constant $c$, then the rank indeed remains at $\Omega(N)$.

Figure 1: VGG19, two class CIFAR, SGD

**Reviewer 2.** We thank Reviewer 2 very much for sharing his/her thoughts on our work. (i) Building new DNNs: We would like to emphasize that the main goal of the work at hand was to deepen the understanding of batch normalization and its crucial interplay with random i.i.d. initialization. Furthermore, as we show in Section 4, rank collapse is not necessarily a problem of the architecture per se but it is closely linked to the way the networks are initialized. As can be seen in Figure 4, a sophisticated (rank preserving) initialization can even outperform BN in very deep networks. However, we do agree that developing new architectures with rank collapse in mind is indeed a very interesting direction to follow up on. As we elaborate next, we think our findings are already of interest to the community, as they offer a novel view on BN which we have not seen in the literature.

(ii) Broadness of impact: We believe that the work at hand makes a significant contribution to a better theoretical understanding of the delicate interplay between architecture, initialization and optimization algorithm. Such a theoretically sound perspective can serve as the basis for further practical advancements on all three of these fronts. One such advancement can be found in Section 4 which includes an accelerating pre-training step that is developed directly from our theoretical analysis. Furthermore, since batch normalization itself is one of the major architectural developments of the last decade with successful application in countless settings, we do believe that this result can be an important step towards a systematic improvement of optimization methods for neural networks based on theoretical intuitions.

**Reviewer 3.** We are very grateful for the comments made by Reviewer 3. (i) $\gamma$: Indeed, our analysis needs $\gamma$ to be small (depending on the width), which comes from the fact that our proof technique relies on low-order Taylor approximations. However, we note that $\gamma$ is, importantly, independent of L! Furthermore, as can be seen in Fig. 3 this seems to be an artefact of our proof technique, since the result holds empirically even for networks without skip connections ($\gamma = \infty$).

(ii) Overfitting: Well spotted! In our examples (Fig. 11) it seems to us that the overfitting tends to happen whenever a certain accuracy is achieved on the training set, regardless of the actual method that is used. Please see Fig. 2 to confirm this intuition: Here, one can see that when reaching $\sim 75\%$ training accuracy (epoch 100 for pre-training and 200 for pre-training) both methods yield a similar test loss ($\sim 1.75$). This is not necessarily surprising as the networks considered are MLPs that might have a harder time at finding generalizable patterns in image data compared to e.g. convnets.

(iii) Extensions to CNNs: This is definitely a very interesting follow up directions. In fact we are currently looking into extending our theoretical result to convolutional neural nets. The notion of rank is a bit more subtle in convolutional layers as the hidden presentation are third order tensors but we still observe that (after unfolding the tensors) randomly initialized convnets suffer from rank collapse unless batch normalization layers are in place (please see Figure 1 above as well as Fig. 11 in the appendix of our paper).

Figure 2: Fig 11 from paper for more epochs

[Meta-Review · NeurIPS 2020]

This paper studies the effect of adding batch normalization (BN) layers on the rank of the data matrix as it passes through the network. The authors prove the rank does not collapse to rank one, which is what occurs when the network does not have BN layers. The theoretical results are accompanied by experiments. Reviewers felt that the paper made an important contribution to understanding models that use BN.